# Intrinsic control of neuronal diversity and synaptic specificity in a proprioceptive circuit

Maggie M Shin[1], Catarina Catela[2], Jeremy Dasen[1]*

[1]Neuroscience Institute, Department of Neuroscience and Physiology, NYU School of Medicine, New York, United States; [2]Department of Neurobiology, University of Chicago, Chicago, United States

**Abstract** Relay of muscle-derived sensory information to the CNS is essential for the execution of motor behavior, but how proprioceptive sensory neurons (pSNs) establish functionally appropriate connections is poorly understood. A prevailing model of sensory-motor circuit assembly is that peripheral, target-derived, cues instruct pSN identities and patterns of intraspinal connectivity. To date no known intrinsic determinants of muscle-specific pSN fates have been described in vertebrates. We show that expression of Hox transcription factors defines pSN subtypes, and these profiles are established independently of limb muscle. The *Hoxc8* gene is expressed by pSNs and motor neurons (MNs) targeting distal forelimb muscles, and sensory-specific depletion of *Hoxc8* in mice disrupts sensory-motor synaptic matching, without affecting pSN survival or muscle targeting. These results indicate that the diversity and central specificity of pSNs and MNs are regulated by a common set of determinants, thus linking early rostrocaudal patterning to the assembly of limb control circuits.

*For correspondence:
Jeremy.Dasen@nyumc.org

**Competing interests:** The authors declare that no competing interests exist.

## Introduction

Sensory-motor circuits within the spinal cord are essential for the coordinated control of limb muscle. In mammals, proprioceptive sensory neurons (pSN) process a continuous stream of data from approximately 50 forelimb or hindlimb muscles, and relay this information to the appropriate circuits tasked to orchestrate motor behaviors (*Imai and Yoshida, 2018*; *Tuthill and Azim, 2018*). The orderly arrangement of connections formed between pSNs and spinal circuits enable animals to seamlessly perform a vast repertoire of limb movements. A single pool of motor neurons (MNs) innervates an individual muscle and receives instructive feedback, not only from sensory neurons, but also descending inputs from supraspinal centers and local spinal interneurons (*Arber, 2012*; *Plant et al., 2018*). These inputs collectively modify the pattern of MN activity thereby choreographing appropriate muscle activation sequences during behavior.

The simplest input pathway to MNs is the monosynaptic reflex arc, composed of a limb muscle innervated by a pool of alpha MNs and type Ia pSNs with stretch-sensing mechanoreceptor endings embedded within muscle spindles. Type Ia pSNs are uniquely poised to provide direct and immediate muscle status information through the monosynaptic connections they establish with MNs. During development, pSN axons navigate through the spinal cord, preferentially contacting MN pools innervating the same peripheral target, while avoiding MNs of functionally antagonist muscles (*Eccles et al., 1957*; *Mears and Frank, 1997*). These connections are remarkably selective, as a single pSN establishes monosynaptic connections with each of the ~50–300 MNs that supply the same muscle target (*Mendell and Henneman, 1968*). While the mechanisms of pSN central specificity are largely unknown, they appear to be established independent of patterned neural activity

(*Mendelsohn et al., 2015*; *Mendelson and Frank, 1991*), suggesting pSN-MN matching relies on genetic programs acting during neural development.

After sensory neurons are born, nascent neurons advance through a hierarchical process of diversification in which expression of specific genes coincides with the acquisition of specialized neuronal characteristics, including peripheral target specificity and central projection pattern (*Dasen, 2009*; *Lallemend and Ernfors, 2012*). Sensory neurons generated at spinal levels derive from neural crest cells which coalesce outside the CNS to form dorsal root ganglia (DRG) (*Butler and Bronner, 2015*). As DRG develop, most sensory neurons co-express the homeodomain transcription factors Isl1 and Brn3a, which are necessary for deployment of pan-sensory neuron genetic programs (*Dykes et al., 2011*). At these early stages, pSNs can be discriminated from other sensory classes by expression of the transcription factors Runx3 and Etv1, the neurotrophin receptor *Nrtk3*, and the calcium binding protein Parvalbumin (PV). Genetic studies in mice indicate that *Runx3* and *Etv1* are essential for establishing and maintaining core features of pSN identity, including their survival and ability to extend central axons into the ventral spinal cord (*Arber et al., 2000*; *Inoue et al., 2002*).

While the transcriptional programs governing features common to all pSNs have been characterized, understanding later developmental facets of sensory neuron specification, such as muscle target specificity and central connectivity, has been particularly challenging. In contrast to the topographic arrangement of spinal MN subtypes, sensory neurons of different modalities and attributes are intermixed within a DRG, with no clear organizational pattern, aside from restricted expression of early determinants involved in establishing a broader sensory neuron class identity (*Honig et al., 1998*; *Jessell et al., 2011*). A dearth of molecular markers for more nuanced neuronal features has made it challenging to characterize how pSNs and other sensory modalities further diversify into specific subtypes. One particular gap in our understanding is how the specificity of central connections between pSNs and MNs of the same muscle is achieved, since pSN axons must distinguish between vast numbers of potential postsynaptic targets within the ventral spinal cord.

A significant contributing factor to the specificity of connections in sensory-motor circuits is the recognition of specific MN subtypes by pSN central afferents. As the limb develops, a network of ~20 Hox transcription factors determines the molecular identities and peripheral target specificities of lateral motor column (LMC) neurons dedicated to limb control (*Philippidou and Dasen, 2013*). Mutation in genes acting downstream of Hox function in MNs, including the transcription factor *Pea3* and synaptic-specificity determinant *Sema3e*, leads to a disruption in the normal pattern of central connections between pSN and MNs (*Fukuhara et al., 2013*; *Pecho-Vrieseling et al., 2009*; *Vrieseling and Arber, 2006*). In addition, genetic transformation of thoracic MNs to a limb-level LMC fate, through mutation in the *Hoxc9* gene, results in the formation of ectopic synaptic connections between limb pSNs and axial muscle-innervating MNs (*Baek et al., 2017*). By contrast, after MN-specific deletion of the *Foxp1* gene, which encodes a factor required for all Hox activity in limb MNs, pSN axons maintain appropriate termination patterns within the ventral spinal cord (*Sürmeli et al., 2011*). However, because MN topographic organization is scrambled in *Foxp1* mutants, limb pSNs form connections with inappropriate MN subtypes. The preservation of pSN central projection pattern in *Foxp1* mutants suggests pSNs acquire specific features that enable them to target specific dorsoventral domains within the spinal cord.

While studies provide evidence for an essential role for MN subtype identity in establishing sensory-motor synaptic specificity, the mechanisms that determine the central pattern of pSN postsynaptic connections are poorly understood. In contrast to MN specification, where key developmental features emerge largely independent of peripheral cues, sensory neuron development relies on extrinsic signals provided by limb mesenchyme and muscle (*Arber, 2012*; *Sharma et al., 2020*; *Wu et al., 2019*). Expression of the *Nrtk3* receptor renders pSNs sensitive to peripheral neurotrophin-3 (*Ntf3*) signaling, and both *Nrtk3* and *Ntf3* are essential for the differentiation and survival of pSNs (*Chen et al., 2003*). *Ntf3/Nrtk3* signaling regulates expression of *Etv1* and *Runx3*, and muscle-by-muscle differences in the level of *Ntf3* expression appear to contribute to pSN subtype diversity (*de Nooij et al., 2013*; *Patel et al., 2003*; *Wang et al., 2019*). Moreover, it has been shown that signals originating from the limb mesenchyme can trigger expression of genes that mark muscle-specific pSN subtypes (*Poliak et al., 2016*). While certain molecular features common to all pSNs have been shown to be limb-independent (*Chen et al., 2002*), whether pSN diversity and synaptic specificity rely on neuronal-intrinsic specification programs remain to be determined. As such there are currently no known fate determinants of muscle-specific pSNs.

We considered the possibility that the same Hox-dependent regulatory networks employed to specify spinal MN subtypes also contribute to the diversification of pSNs during sensory-motor circuit assembly. We show that selective expression of Hox proteins defines pSN populations generated at specific rostrocaudal levels, paralleling Hox expression in spinal MNs. Expression of *Hox* genes is maintained in both pSNs and MNs after removal of the developing limb bud, indicating that neuronal *Hox* pattern is initially established independent of target-derived cues. We found that distal forelimb flexor muscles, and the MNs that innervate them, are targeted by pSNs expressing the *Hoxc8* gene. In the absence of *Hoxc8* function, forelimb pSNs establish ectopic monosynaptic contacts on MNs innervating functionally antagonist forelimb muscles. These studies provide evidence for a neuronal-intrinsic program in which the selective activities of Hox proteins encodes key features of pSN diversification and target selectivity.

## Results

### *Hox* expression delineates subpopulations of pSNs along the rostrocaudal axis

To explore a potential role for *Hox* genes in pSN diversification, we analyzed the expression of individual Hox proteins in spinal DRG during the early phases of sensory neuron development (*Figure 1*). We examined Hox protein expression in relation to Runx3, Etv1, and PV, three markers predominately restricted to pSN subtypes. Because the patterns of *Hox* gene expression within the spinal cord are most thoroughly characterized in cervical (C) segments (*Catela et al., 2016*; *Lacombe et al., 2013*), we focused on the pattern of *Hox* expression in DRG generated between C2-C8. We began by analyzing the DRG expression of a subset of *Hox4-Hox8* paralog genes between E12.5-E14.5, stages in which pSN axons have reached their muscle targets and central afferents have begun to invade the dorsal spinal cord (*Hippenmeyer et al., 2002*; *Kramer et al., 2006*). We found that subpopulations of cervical DRG neurons selectively coexpressed Hox proteins and molecular markers of pSN identity (*Figure 1*, *Table 1*, *Figure 1—figure supplement 1*). Hox proteins expressed by pSNs included Hoxc4, Hoxa5, Hoxc6, Hoxa7, and Hoxc8, which also collectively define forelimb LMC neuron diversity (*Figure 1a–e*; *Dasen et al., 2005*). Each of these Hox proteins were detected at cervical levels and/or rostral thoracic segments, but were not present in caudal thoracic or lumbar DRG (data not shown). Within individual cervical DRG, Hox proteins were expressed by a subset of pSNs, and the fraction of pSNs expressing a given *Hox* gene within a single DRG varied along the rostrocaudal axis (*Figure 1—figure supplement 1a*). Interestingly, members of the *Hoxb* gene cluster, including *Hoxb4*, were also restricted to cervical segments, but appeared to be more broadly expressed by DRG classes (*Figure 1—figure supplement 1b*, data not shown). These observations indicate that members of the *Hoxa* and *Hoxc* gene clusters are expressed by subsets of cervical pSNs.

Within a single DRG, a proportion of pSNs also demonstrated co-expression of multiple Hox proteins. For example, within individual cervical DRG, a subset of Hoxc8$^+$ cells coexpressed Hoxa7, a subset of Hoxa5$^+$ pSNs co-expressed Hoxc4, and a subset of Hoxc6$^+$ pSNs expressed Hoxc8 (*Figure 1f,g*, *Figure 1—figure supplement 1c*). Furthermore, DNA-binding cofactors known to be essential for Hox activity (*Merabet and Mann, 2016*), including Meis2 and Pbx3, were detected in pSNs (*Figure 1h*, *Figure 1—figure supplement 1d*). Both Meis2 and Pbx3 were expressed by pSNs but also observed in non-proprioceptive sensory neuron subtypes, and lacked rostrocaudal specificity. These observations suggest that the combinatorial actions of Hox proteins and their co-factors could account for subtype diversity of cervical pSNs.

We next compared the expression of individual *Hox* genes in sensory neurons along the rostrocaudal axis of the spinal cord. While certain *Hox* genes are coexpressed within the same sensory neuron, others demonstrate clear boundaries from one another and do not co-localize, despite being expressed in the same sensory class. For example, Hoxa5 expression was confined to pSNs in rostral cervical segments (C2-C5) while Hoxc8 expression was restricted to caudal cervical and rostral thoracic DRG (C6 to T2) (*Figure 1i,j*). Thus, Hoxa5 and Hoxc8 expression by pSNs is mutually exclusive and mirrors the restricted expression pattern of Hoxa5/Hoxc8 in forelimb-innervating MNs. These observations indicate that pSN subtypes can be delineated by differential *Hox* gene

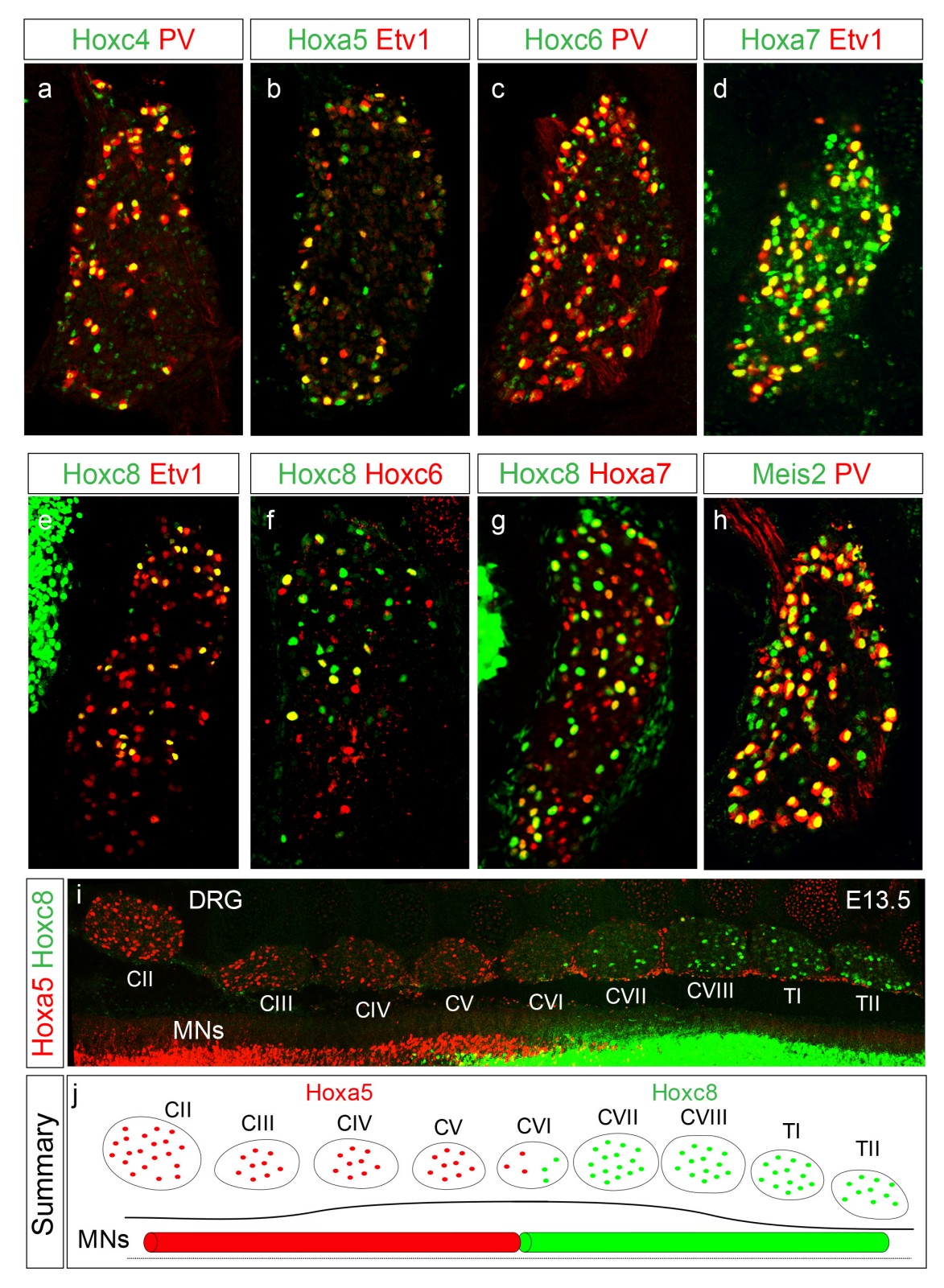

**Figure 1.** Restricted *Hox* expression defines pSN subtype identity. (**a–e**) Expression of indicated Hox proteins in cervical DRG. Images show cross sections of DRG located between segmental levels C2-C8 of E14.5 mouse embryos stained with Hox and the pSN markers PV or Etv1. Images are representative of at least three embryos analyzed at this age. (**f**) Coexpression of Hoxc6 and Hoxc8 in subsets of pSNs in caudal cervical DRG. (**g**) Coexpression of Hoxc8 and Hoxa7 in sensory neurons. (**h**) Meis2 coexpression with PV in sensory neurons. (**i**) Top-down view of E13.5 mouse spinal cord

*Figure 1 continued on next page*

*Figure 1 continued*

showing mutually exclusive expression of Hoxa5 and Hoxc8 in DRG. Spinal MNs also abide by restricted rostrocaudal domains of Hoxa5 and Hoxc8 expression. (j) Summary of Hoxa5 and Hoxc8 expression pattern in MNs and pSNs. See also *Figure 1—figure supplement 1*.

The online version of this article includes the following figure supplement(s) for figure 1:

**Figure supplement 1.** Hox expression pattern of sensory neurons in cervical DRG.

expression, and suggest that pSNs employ early rostrocaudal patterning mechanisms similar to those of MNs.

## Neuronal expression of *Hox* genes is initially limb-independent

During the early stages of neural tube development, expression of *Hox* genes is initiated by secreted morphogens acting on progenitors along the rostrocaudal axis (*Bel-Vialar et al., 2002*; *Dasen et al., 2003*; *Liu et al., 2001*). These early patterning signals induce *Hox* expression in the neural tube prior to limb bud formation. By contrast, studies of limb and non-limb innervating pSNs have shown that the molecular identities and central projection patterns of pSNs are established and maintained through extrinsic, target-derived, signals (*de Nooij et al., 2013*; *Poliak et al., 2016*). These findings raise the question of what the relative contributions of early patterning signals and target-derived cues are in regulating *Hox* expression in pSNs.

To answer this question, we used limb-bud ablation assays in chick embryos to determine whether *Hox* expression in pSNs persists after removal of signals provided by limb mesenchyme and muscle. We first examined whether the expression of Hox proteins in sensory neurons is conserved between mouse and chick. We found that Hoxa5, Hoxc6, and Hoxc8 were selectively expressed by forelimb-level DRG by st31 (equivalent to E13.5 in mouse) (*Figure 2a,b*). As in mouse DRG, Hoxa5 was expressed by rostral cervical DRG, Hoxc8 was expressed by caudal cervical sensory neurons, while Hoxc6 was expressed in both rostral and caudal cervical DRG (*Figure 2a,b*). Co-staining with the pSN-restricted marker Runx3 indicated that Hox proteins are expressed by pSNs in chick, with some notable differences from mouse. In chick, Hoxa5 was broadly expressed by rostral cervical sensory neurons, while Hoxc8 was detected in a smaller fraction of caudal cervical pSNs (*Figure 2—*

**Table 1.** Expression of *Hox4-Hox8* paralog proteins in SNs.

Table lists each of the tetrapod *Hox4-Hox8* gene paralogs and their expression pattern in SNs between segmental levels CII-TII. Not all antibody combinations were tested for Hox co-expression. ND, not detected; NT, not tested.

| Hox protein | DRG expression | pSN expression | RC level | Hox co-expression |
|---|---|---|---|---|
| Hoxc4 | yes | yes | CII-CVII | Hoxa5 |
| Hoxc5 | NT | - | - | - |
| Hoxc6 | yes | yes | CIV-CVIII | Hoxc8 |
| Hoxc8 | yes | yes | CVI-TII | Hoxc6, Hoxa7 |
| Hoxa4 | NT | - | - | - |
| Hoxa5 | yes | yes | CII-CVI | Hoxc4 |
| Hoxa6 | ND | - | - | - |
| Hoxa7 | yes | yes | CV-TII | Hoxc8 |
| Hoxb4 | yes | no | NT | - |
| Hoxb5 | yes | no | NT | - |
| Hoxb6 | NT | - | - | - |
| Hoxb7 | NT | - | - | - |
| Hoxb8 | NT | - | - | - |
| Hoxd4 | NT | - | - | - |

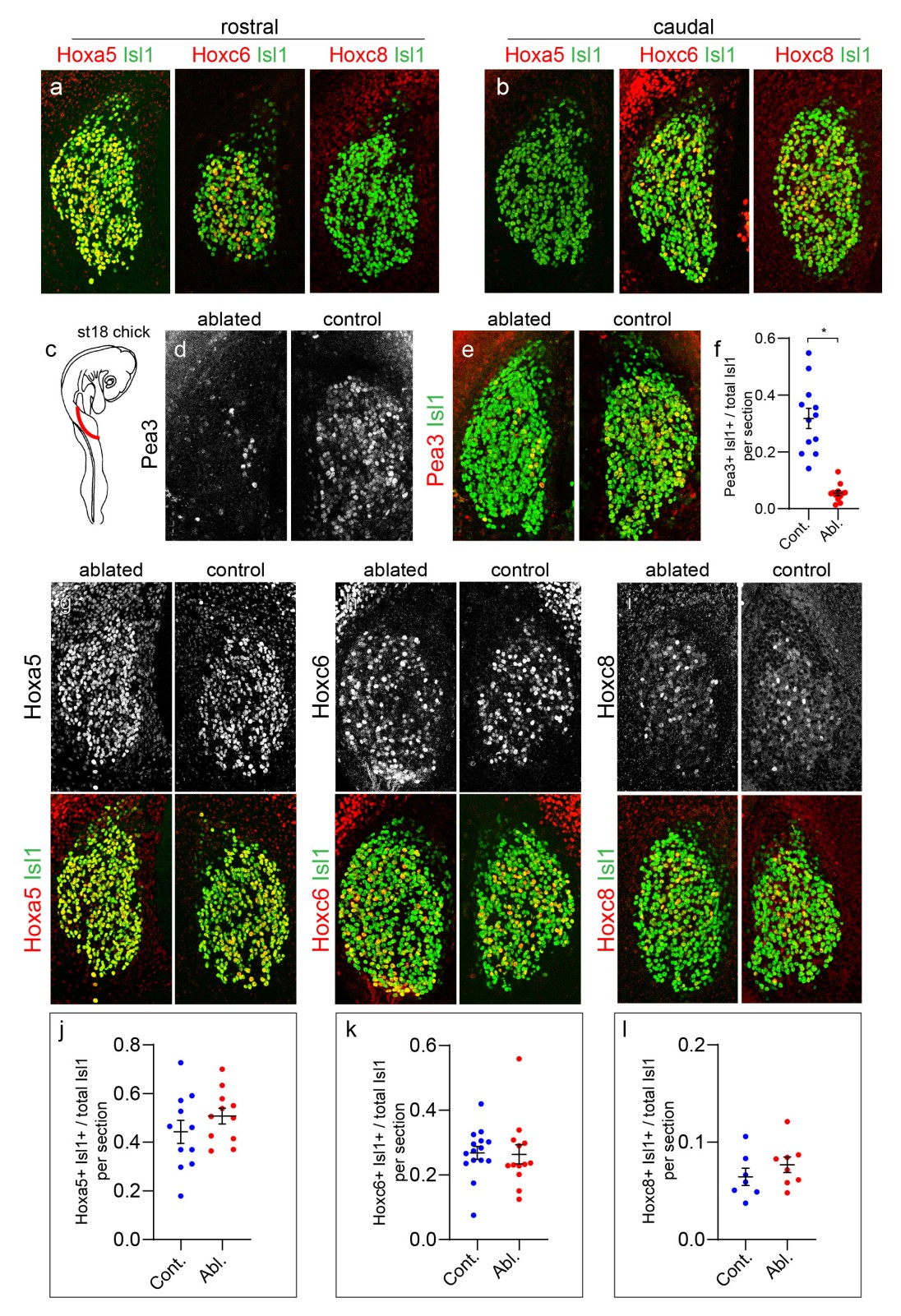

**Figure 2.** Limb-independent expression of *Hox* genes in sensory neurons. (**a and b**) Expression of Hoxa5, Hoxc6, and Hoxc8 in chick forelimb level DRG shown with pan-sensory neuron marker Islet1 (Isl1). Images show cross sections of chick DRG at ~St 26. Hoxa5 and Hoxc6 are present at rostral cervical segments while Hoxc8 is absent from this region (**a**). Hoxa5 is not found in caudal cervical DRG whereas Hoxc6 and Hoxc8 are present within this region (**b**). Images are representative of at least three embryos analyzed at this age. (**c**) Unilateral forelimb bud ablation of chick at St 16–18. Embryos

*Figure 2 continued on next page*

*Figure 2 continued*

harvested at St 26–28. (**d–e**) Loss of Pea3 expression on the limb-ablated side in relation to the non-ablated side. Expression of Isl1 is unaffected at this stage. Images show cervical DRG of an individual embryo at the same segmental levels between ablated and non-ablated sides for each panel. (**f**) Quantification of loss of Pea3, as a fraction of total Isl1$^+$ SNs. Controls, 31.8 ± 3.6%, N = 12 sections from three animals, ablated 5.5 ± 0.1%, N = 12 sections from three animals, p<0.0001, Student's t-test. (**g–i**) Top panels show expression of individual Hox expression. Bottom panels show Hox expression with Isl1. There is no difference in Hox expression between the ablated and non-ablated side for Hoxa5$^+$ SNs in rostral cervical segments (**g**), Hoxc6$^+$ SNs in rostral and caudal cervical segments (**h**), or Hoxc8$^+$ SNs in caudal cervical segments (**i**). (**j–k**) Quantification of fraction of Hox$^+$Isl1$^+$ over total Isl1$^+$ SNs in control and limb-ablated chick embryos. For each Hox protein, sections were obtained from three limb-ablated embryos, with non-ablated side of embryo serving as the control. Hoxa5 (44.3 ± 4.7% in N = 11 control sections, 50.7 ± 3.2% in N = 11 limb-ablated sections, p=0.27, Student's t-test), Hoxc6 (26.8 ± 2.0% in N = 15 control sections, 26.3 ± 3.0% in N = 13 limb-ablated sections, p=0.90), and Hoxc8 (6.4 ± 0.9% in N = 7 control sections, 7.6 ± 0.8% in N = 8 sections ablated, p=0.32). See also *Figure 2—figure supplement 1*.

The online version of this article includes the following source data and figure supplement(s) for figure 2:

**Source data 1.** Quantification of markers proteins after limb bud ablation.
**Figure supplement 1.** Hox expression in MNs and SNs is limb-independent.

*figure supplement 1a–c*). The reduced number of Hoxc8$^+$ pSNs in chick versus mouse likely reflect evolutionary changes in the distribution and function of avian and rodent forelimb muscle.

We next unilaterally ablated the forelimb bud of chick embryos at stage (St) 16–18, a phase when the initial rostrocaudal profiles of *Hox* gene expression have been established, but prior to the appearance of postmitotic pSNs and MNs (~E8.5 in mouse) (*Figure 2c*). After limb bud extirpation, embryos were allowed to continue to develop for 3 days (to ~st26-28 [~E11.5-12.5 in mouse]). At this age, all MNs and pSNs have been generated, but have not reached the phase where they rely on limb-derived neurotrophic support. To confirm successful removal of limb-derived cues, we examined expression of the ETS protein Pea3, which is expressed by subsets of pSNs and MNs in a limb-dependent manner (*Lin et al., 1998*). After forelimb bud ablation, the number of sensory neurons and MNs expressing Pea3 markedly decreased relative to the non-ablated side (*Figure 2e*, *Figure 2—figure supplement 1e*). The fraction of Isl1$^+$ SNs expressing Pea3 was reduced to 5.5 ± 0.1% (mean ± SEM), compared to 31.8 ± 3.6% in controls (p<0.0001, Student's t-test) (*Figure 2f*). In addition, the number of SNs expressing Runx3 was reduced from 15.5 ± 2.4% in controls to 7.3 ± 0.9% after limb ablation (p=0.0038, Student's t-test) (*Figure 2—figure supplement 1d*). The decrease in Pea3 expression was not a result of the general loss of sensory neurons as the number of Isl1$^+$ DRG neurons, a pan-sensory neuron marker, was comparable between the ipsilateral and contralateral sides of the ablated limb (*Figure 2e*).

In contrast to the loss of Pea3, expression of Hoxa5, Hoxc6, and Hoxc8 were unchanged in both sensory neurons and MNs after forelimb removal (*Figure 2g–i*, *Figure 2—figure supplement 1a–c, e–g*). The fraction of Isl1$^+$ SNs expressing Hoxa5 (44.3 ± 4.7% in controls, 50.7 ± 3.2% ablated, p=0.27, Student's t-test), Hoxc6 (26.8 ± 2.0% controls, 26.3 ± 3.0% ablated, p=0.90), and Hoxc8 (6.4 ± 0.9% controls, 7.6 ± 0.8% ablated, p=0.32) was not significantly changed (*Figure 2j,k,l*). Because expression of the pSN markers Runx3 and Etv1 are reduced after limb ablation, we were unable to quantify the fraction of pSNs that retain Hox expression. Nevertheless, these results are consistent with a model in which the pattern of *Hox* gene expression in sensory neurons and MNs is initiated through intrinsic genetic programs that operate independent of limb-derived cues.

## Hoxc8 expression in type-Ia pSNs during sensory-motor circuit maturation

To further examine contribution of *Hox* genes to the diversification of sensory neuron subtypes, we performed a detailed characterization of Hox protein expression in relation to the ontogeny of sensory-motor circuit development (*Figure 3*). We focused our studies on *Hoxc8* for this analysis, due to its central role in establishing the molecular identities and muscle-target specificity of MNs innervating distal forelimb muscles of mouse and chick (*Catela et al., 2016*; *Dasen et al., 2005*).

To determine at which phase of sensory-motor circuit development *Hoxc8* might be required, we analyzed the ontogeny of Hoxc8 protein expression in pSNs in mouse. Since the levels of Hoxc8 protein expression in the CNS attenuate at later stages of embryonic development, we utilized a conditional *Hoxc8* allele in which a *LacZ* reporter is expressed upon Cre-dependent excision of *Hoxc8* coding sequence (*Blackburn et al., 2009*; *Catela et al., 2016*). To achieve sensory neuron-restricted

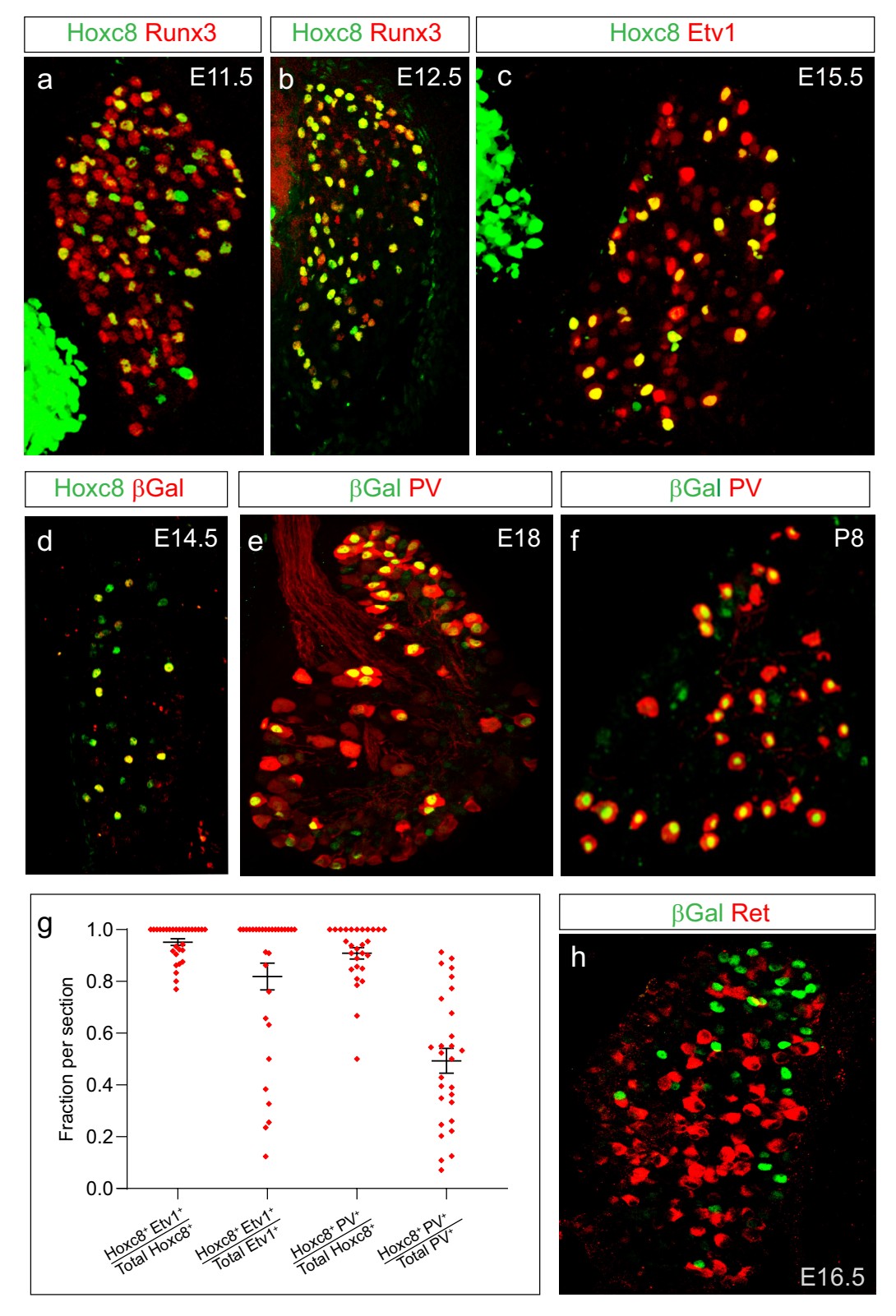

**Figure 3.** Profile of *Hoxc8* in pSNs during sensory-motor circuit development. (**a, b**) Expression of Hoxc8 with the pSN marker Runx3 at E11.5 (**a**) and E12.5 (**b**). (**c**) Hoxc8 and Etv1 expression in pSNs at E15.5. (**d**) Colocalization of Hoxc8 and βGal in *PLAT::Cre; Hoxc8^LacZ-flox/+* mice at E14.5. (**e, f**) Colocalization of βGal with the pSN marker PV at E18 (**e**) and P8 (**f**). (**g**) Quantification of Hoxc8 with either Etv1 or PV at E14.5. Each data point shows individual sections taken from at least three mice. Hoxc8$^+$Etv1$^+$/total Hoxc8$^+$ cells = 0.95 ± 0.01 (mean ± SEM, N = 30 sections); Hoxc8$^+$Etv1$^+$/total

*Figure 3 continued on next page*

*Figure 3 continued*

Etv1[+] cells = 0.82 ± 0.01 (N = 30 sections); Hoxc8[+]PV[+]/total Hoxc8[+] cells = 0.91 ± 0.02 (N = 28 sections); Hoxc8[+]PV[+]/total PV[+] cells = 0.50 ± 0.04 (N = 28 sections). (h) Non-overlapping expression of βGal and Ret, a marker for subpopulations of cutaneous sensory neurons at E16.5. See also *Figure 3—figure supplement 1*.

The online version of this article includes the following source data and figure supplement(s) for figure 3:

**Source data 1.** Quanification of marker proteins in sensory neurons.

**Figure supplement 1.** Detailed expression profile of *Hoxc8*.

*LacZ* reporter expression, we crossed *Hoxc8^{LacZ-flox/+}* mice to an *PLAT::Cre* line. This line drives Cre expression in the neural crest cells from which spinal sensory neurons are derived, but is excluded from neurons in the central nervous system (*Pietri et al., 2003*). This breeding strategy allowed us to unambiguously identify Hoxc8[+] pSNs at later postnatal stages, due to the persistence of βGal protein expression. At E14.5-E15.5 all βGal positive cells expressed Hoxc8 and Runx3 proteins (*Figure 3d*, *Figure 3—figure supplement 1d,e*), indicating this strategy recapitulates the normal pattern of *Hoxc8* expression in pSNs.

Presumptive pSNs initiate Hoxc8 protein expression at E11.5, shortly after the appearance of postmitotic sensory neurons (*Figure 3a*). Hoxc8 is maintained between E12.5 and E15.5, the time window in which pSNs extend axons to their peripheral muscle targets and central afferents begin to enter the dorsal spinal cord (*Figure 3b,c*). Hoxc8 and βGal continued to be expressed through the first postnatal week, during the phase in which pSNs extend central projections within the ventral spinal cord and connect with central postsynaptic targets (*Figure 3e,f*, *Figure 3—figure supplement 1*). Therefore, based on our analysis, expression of Hoxc8 coincides with the period of embryonic development when forelimb-innervating pSNs begin forming connections with their peripheral muscle and central postsynaptic targets.

To further confirm the specificity of Hoxc8 in pSNs we examined its expression in relation to Etv1 and PV between segmental levels C6-T1 at E14.5. The majority (>90%) of Hoxc8[+] neurons expressed Etv1 and PV, consistent with a pSN-restricted expression pattern (*Figure 3g*, *Figure 3—figure supplement 1a,b*). At these segmental levels, however, Hoxc8 was expressed by only ~50% of the total PV[+] population (*Figure 3g*). Between segments C6-C8, 15–20% of PV[+] sensory neurons have been shown to co-express Ret, a marker for a subset of cutaneous sensory neurons (*Niu et al., 2013*). We found that all Hoxc8[+] sensory neurons lacked Ret expression. These results indicate a fraction of cervical pSNs express Hoxc8, but that cervical PV[+]Ret[+] sensory neurons, likely cutaneous sensory neurons, are Hoxc8[-] (*Figure 3g,h*).

## Peripheral muscle target specificity of Hoxc8[+] pSNs

In spinal MNs, the profile of Hox expression along the rostrocaudal axis is correlated with the position of muscles along the proximal-distal and anterior-posterior axes of the limb. Rostral cervical Hoxa5[+] LMC neurons typically innervate more anterior/proximal forelimb muscles, while caudal cervical Hoxc8[+] MNs project to distally and/or posteriorly located forelimb muscles (*Catela et al., 2016*). Because the rostrocaudal profile of *Hox* genes in pSNs mirrors that of spinal MNs, and Hoxc8[+] MNs are known to innervate distal forelimb muscles, we sought to evaluate muscle target selectivity of Hoxc8[+] pSNs.

To identify the peripheral muscle targets of Hoxc8[+] pSNs, we labeled pSNs through intramuscular injection of Cholera toxin B subunit (CTB) and examined Hoxc8/βGal protein expression in retrogradely labeled neurons. We performed retrograde tracing assays on nine specific forelimb muscles, ranging from proximal to distal positions, and varying in size, but sharing a common role in controlling arm, wrist, or digit movement (*Figure 4a*). In the distal forelimb, flexor muscles are positioned ventrally and act to adduct the wrist and flex the digits. Conversely, distal extensor muscles reside dorsally and act as antagonists to distal flexors. Although each of the major forelimb-controlling muscles were injected and processed for analysis, a few were excluded due to inaccessibility, as deeper muscles would require the removal of the overlying musculature. CTB was injected into single forelimb muscles of wildtype and *PLAT::Cre; Hoxc8^{LacZ-flox/+}* mice at P4, thereby retrogradely labeling sensory neuron afferent fibers that have taken up CTB tracer through direct contact with muscle (N ≥ 3 animals/muscle). Spinal cords with attached DRG were then isolated at P7 to assess

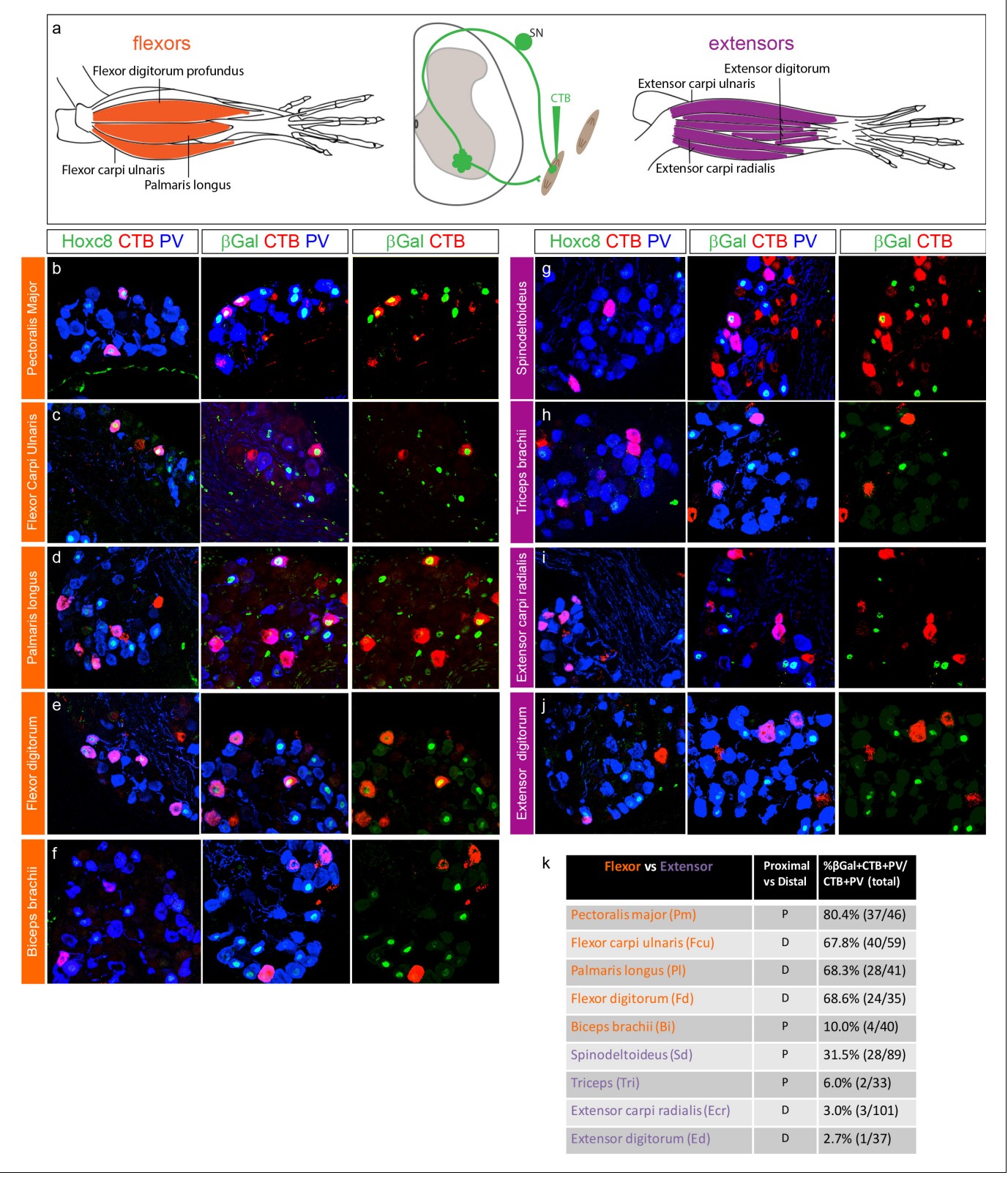

**Figure 4.** Hoxc8 pSNs preferentially target distal flexor limb muscles. (**a**) Schematic of distal forelimb muscles of mouse. Limb flexor muscles shown in orange, extensors in purple. Muscles were injected with CTB at P4 and collected at P7. (**b–j**) Cross sections of caudal cervical DRG of *PLAT::Cre; Hoxc8^LacZ-flox/+* mice injected with CTB in an individual forelimb muscle. First column shows colocalization of Hoxc8, CTB, and PV; second column colocalization of βGal, CTB, and PV; and third column βGal and CTB. (**k**) Table of injected forelimb muscles with quantification of βGal+ pSN

*Figure 4 continued*

innervation shown as percentage (βGal⁺CTB ⁺ PV⁺ SNs over total CTB⁺PV⁺SNs in all sections) with raw numbers next to percentages. Values in table show cumulative data from CTB-labeled DRG sections collected from at least three mice per muscle. See also *Figure 4—figure supplement 1*.

The online version of this article includes the following figure supplement(s) for figure 4:

**Figure supplement 1.** Characterization of Hoxc8⁺ pSN innervation of forelimb muscles.

representative populations of sensory neurons innervating the injected muscle. Injections were performed no earlier than P4 due to the thin size of the distal forelimb muscles as well as the inefficiency of neonatal CTB labeling, a probable outcome of lower expression levels of the CTB receptor in neonates (*Yu, 1994*). The coincidence of CTB/Hoxc8/PV and CTB/βGal/PV labeling was then analyzed in DRG to determine if the muscle received innervation from Hoxc8⁺ pSNs.

With the exception of the biceps brachii, all of the injected flexor muscles were found to be innervated by predominately Hoxc8⁺ pSNs, including the pectoralis major (PM), flexor carpi ulnaris (FCU), palmaris longus (PL), and flexor digitorum profundus (FDP). For the PM, FCU, PL, and FDP,~70% or more of the total CTB labeled pSNs were Hoxc8⁺ (*Figure 4f–f,k*, *Figure 4—figure supplement 1a–e*). The biceps brachii is proximally located in relation to the limb, and innervated by Hox5⁺ MNs, while the latter three flexors inhabit the distal forelimb and are supplied by Hoxc8⁺ median and ulnar MNs (*Catela et al., 2016*). The PM is also considered a proximal forelimb muscle, though it is one of the largest arm flexion-controlling muscles responsible for a wide range of arm movements. Of the injected extensor muscles, including the proximally located triceps (Tri) and distally positioned extensor carpi radialis (ECR) and extensor digitorum (ED), a small to negligible percentage (3–6%) of CTB labeled pSNs expressed Hoxc8 (*Figure 4g–k*, *Figure 4—figure supplement 1f–I*). After injection into the spinodeltoideus, 32% of labeled neurons expressed Hoxc8, possibly reflecting innervation by pSNs with a mixed molecular profile. These results indicate that Hoxc8⁺ pSNs preferentially target muscles involved in distal forelimb flexion.

## Sensory neuron survival and differentiation in *Hoxc8^{SNΔ}* mice

We next evaluated the function of *Hoxc8* during pSN development by generating homozygous *Hoxc8^{LacZ-flox/LacZ-flox}* mice expressing *PLAT::Cre* (referred to henceforth as *Hoxc8^{SNΔ}* mice). In *Hoxc8^{SNΔ}* animals, expression of Hoxc8 protein is selectively removed from SNs but maintained by neurons within the spinal cord (*Figure 5—figure supplement 1a*). Because *Hoxc8* has been shown to be essential for the survival of a subset of caudal cervical LMC neurons after E12.5 (*Catela et al., 2016*; *Tiret et al., 1998*), this posed the possibility that *Hoxc8* is similarly involved in the selective survival or maintenance of caudal cervical pSNs during embryogenesis. Alternatively, since Hoxc8 expression persists through the 1ˢᵗ postnatal week in sensory neurons this suggests a potential function in later aspects of pSN maturation and connectivity.

We therefore examined the function of *Hoxc8* during midgestation (E14.5-E15.5) and postnatally (P4-P7). To clearly visualize Hoxc8⁺ populations at postnatal stages, we used the inserted *LacZ* reporter which expresses βGal in lieu of *Hoxc8*, enabling us to track the fate of pSNs lacking *Hoxc8*. We compared the number of βGal⁺ cells between *Hoxc8^{LacZ/+}* and *Hoxc8^{SNΔ}* animals in DRG C8, where a subset of Hoxc8⁺ pSNs reside. At P7, the percentage of PV⁺ pSNs that expressed βGal was similar between control and *Hoxc8^{SNΔ}* animals (42 ± 3% in N = 33 sections from three control mice, versus 45 ± 3% in N = 32 section from 3 *Hoxc8^{SNΔ}* mice, p=0.43, Student's t test) (*Figure 5a,c*). We also compared the fraction of PV⁺ pSNs that expressed Isl1, which was also unchanged (22 ± 1% in N = 68 sections from three control animals, versus 23 ± 1% in N = 74 section from 3 *Hoxc8^{SNΔ}* mice, p=0.59, Student's t test) (*Figure 5b,c*). Moreover, the distribution of sensory neurons expressing Etv1, PV and Isl1 was grossly unchanged in *Hoxc8^{SNΔ}* animals at E15.5 compared to that of control animals (*Figure 5—figure supplement 1e,f*). All βGal⁺ cells also lacked Ret expression in *Hoxc8^{SNΔ}* animals, indicating that their fate had not been switched to that of PV⁺ cutaneous sensory neurons (*Figure 5—figure supplement 1g,h*). These observations indicate that *Hoxc8* is not required for the survival or maintenance of early pSN molecular features.

To determine if *Hoxc8* is necessary for the ability of cervical pSNs to innervate their normal forelimb muscle targets, we examined the formation of muscle spindles in the palmaris longus (PL) and flexor carpi ulnaris (FCU), two distal forelimb flexor muscles that normally receive input from Hoxc8⁺

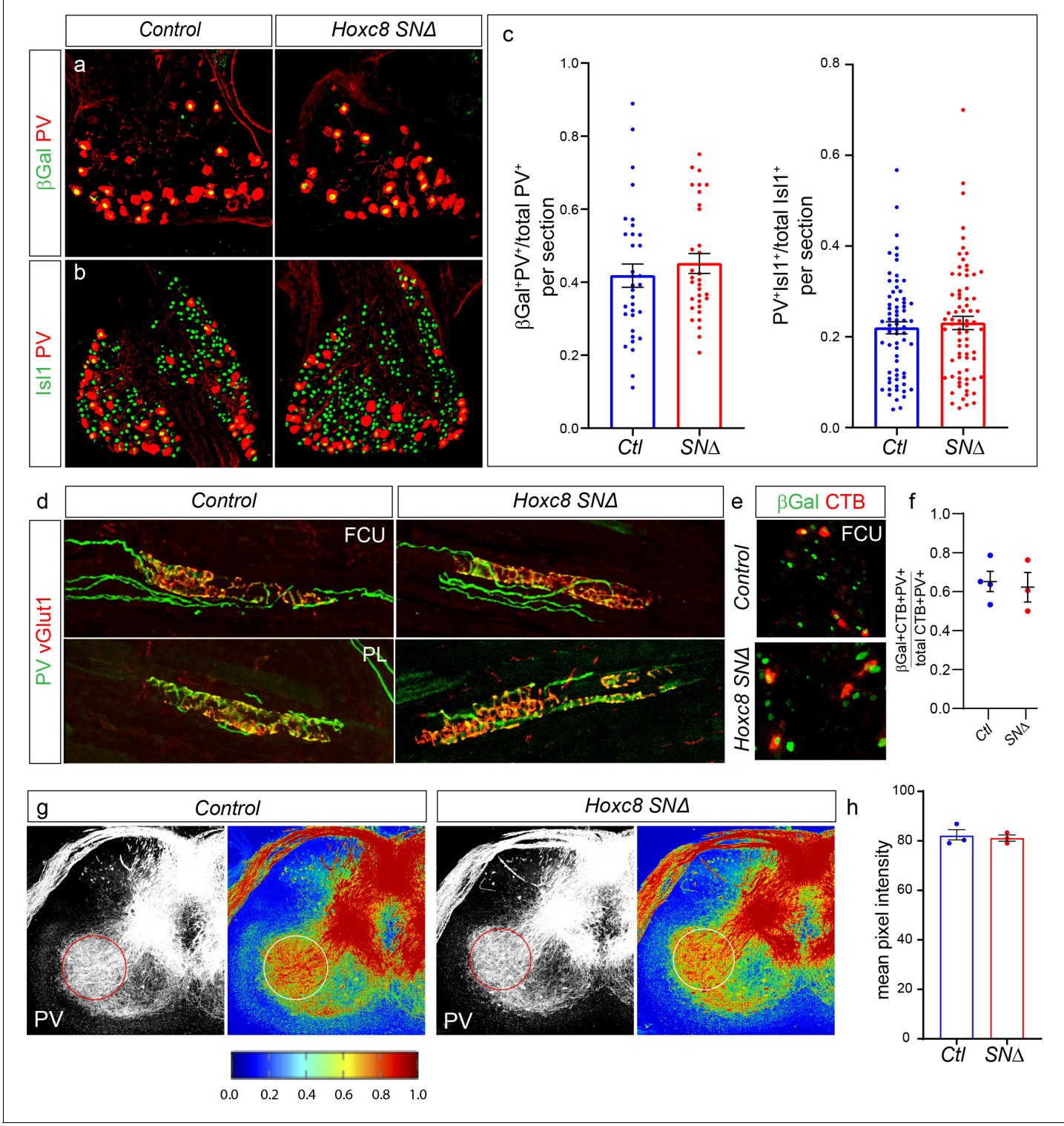

**Figure 5.** *Hoxc8* is dispensable for pSN survival and differentiation. (**a**, **b**) Example images showing expression of βGal and PV (**a**) or PV and Isl1 (**b**) in controls and *Hoxc8$^{SN\Delta}$* mice at P7. (**c**) Quantification of fraction of βGal$^+$ PV$^+$ SNs over total PV$^+$ population: 42 ± 3% (mean ± SEM) in N = 33 sections from three mice, versus 45 ± 3% in N = 32 sections from 3 *Hoxc8$^{SN\Delta}$* mice (p=0.43, Student's t test). Quantification of PV$^+$ Isl1$^+$ SNs over total Isl1$^+$ population: 22 ± 1% in N = 68 sections from three control mice, versus 23 ± 1% for N = 74 sections from 4 *Hoxc8$^{SN\Delta}$* mice (p=0.59, Student's t test). Data points in graphs show results from individual sections. Lines in graph show mean ± SEM. (**d**) Distal forelimb flexors FCU and PL still receive normal pSN innervation and muscle spindles develop normally in *Hoxc8$^{SN\Delta}$* mice compared to controls shown with PV and vGlut1 staining. (**e**) Retrograde labeling of βGal$^+$ SNs after CTB injection of FCU in control and *Hoxc8$^{SN\Delta}$* mice. Muscle injection performed at ~P4 and DRG collected at ~P7 P8. (**f**)

*Figure 5 continued on next page*

Figure 5 continued

Quantification of CTB$^+$βGal$^+$PV$^+$/total CTB$^+$PV$^+$ SNs after FCU retrograde tracing: 65.2 ± 5.1% for N = 4 control mice; 62.3 ± 7.6% for N = 3 Hoxc8$^{SNΔ}$ mice, p=0.75, Student's t test. (g) No difference in PV fiber density in the ventral spinal cord between control and Hoxc8$^{SNΔ}$ mice. PV fiber stain with heat map below. (h) Quantification of the average PV pixel intensity at DRG C8 level. PV fiber density calculated only in ROI created in ventral spinal cord region. Lines indicate mean ± SEM. Average intensity for control: 82.1 ± 2.4, N = 3 mice. Average intensity for Hoxc8$^{SNΔ}$: 81.1 ± 1.3, N = 3 mice (p=0.74, Student's t test). See also *Figure 5—figure supplement 1*.

The online version of this article includes the following source data and figure supplement(s) for figure 5:

**Source data 1.** Quantification of sensory markers in control and Hoxc8 mutants.
**Figure supplement 1.** Preservation of sensory neuron identities in Hoxc8$^{SNΔ}$ mice.

pSNs. We found no discernible difference in the pattern of PV or vGlut1, which accumulate on the peripheral terminals of pSNs, indicating that PL and FCU muscle connectivity is unaltered in the absence of *Hoxc8* (*Figure 5d*). We also tested whether the FCU receives innervation from appropriate pSN subtypes in the absence of *Hoxc8*. We injected CTB into the FCU of Hoxc8$^{SNΔ}$ mice at P4 and collected spinal cords with attached DRG at P7. We found that the fraction of pSNs that were βGal$^+$ CTB$^+$ was similar between controls and Hoxc8$^{SNΔ}$ mice (65.2 ± 5.1% for N = 4 controls; 62.3 ± 7.6% for N = 3 Hoxc8$^{SNΔ}$ mice, p=0.75, Student's t test) (*Figure 5e,f*). Loss of *Hoxc8* therefore does not preclude the ability of cervical pSNs to reach their normal muscle targets, demonstrating that *Hoxc8* is not essential for pSN peripheral projection and target specificity.

Genetic ablation of early pSN fate determinants, including *Runx3* or *Etv1*, leads to marked reduction in the extension of pSN central afferents into the ventral spinal cord (*Arber et al., 2000*; *Inoue et al., 2002*). Since deletion of *Hoxc8* does not affect pSN survival or peripheral innervation, we next asked whether *Hoxc8* is required for the ventral extension of pSNs towards MNs. Hoxc8$^+$ pSN projections originating from DRG C8 terminate within the ventral spinal cord predominantly at this same segmental level (*Baek et al., 2017*). Thus, a noticeable loss of projections to the ventral spinal cord would be evident at this segmental position. We used PV labeling to measure the density of pSN collateral projections terminating in the ventrolateral area of the spinal cord. We observed no difference in the mean pixel intensity of PV fibers innervating the region occupied by forelimb MNs between Hoxc8$^{SNΔ}$ mice and controls (82.1 ± 2.4 for N = 3 controls; 81.1 ± 1.3 for N = 3 Hoxc8$^{SNΔ}$ mice, p=0.74, Student's t test) (*Figure 5g,h*). Collectively, these results indicate that *Hoxc8* is dispensable for pSN survival, peripheral muscle target selection, and ability of pSNs to extend central axons ventrally.

## Altered topography of pSN central connections in *Hoxc8$^{SNΔ}$* mice

We next considered the possibility that deletion of *Hoxc8* disrupts the pattern of central connectivity between muscle-specific pSNs and MNs. To examine pSN synaptic specificity, we employed a modified rabies labeling strategy which directs monosynaptically-restricted anterograde transfer of virus from pSNs to neurons within the spinal cord (*Zampieri et al., 2014*). We bred Hoxc8$^{SNΔ}$ mice with a Cre-dependent line (*Gt(ROSA)26Sor*$^{CAG-loxp-STOP-loxp-rabies-G-IRES-TVA}$ mice, henceforth referred to as *RGT*) expressing two rabies helper proteins: TVA, an avian-specific receptor protein, which permits infection to rabies virus pseudotyped with EnvA, and a rabies glycoprotein, which allows transsynaptic transfer of the virus, both produced in sensory neurons following recombination using the *PLAT:: Cre* line (*Figure 6a*). The injected RVΔG-mCherry-EnvA (RabV) virus lacks its own glycoprotein rendering it incapable of spreading in the absence of the supplanted source. Sensory-restricted Cre expression confines the spread of mCherry-expressing rabies from the injected muscle to the connected pSNs, and subsequently their monosynaptically-coupled postsynaptic partners, while preventing infection of MNs directly from the muscle. An advantage of utilizing this method is that the RabV labels the entire soma of infected neurons making it relatively easy to identify coupled MNs.

We tested the specificity of this tracing assay by injecting distal flexor muscles with RabV in both Cre$^+$ and Cre$^-$ *RGT* mice (*Figure 6—figure supplement 1a–d*). In *PLAT::Cre$^+$ RGT* mice, RabV injected into flexor muscles labeled the connected pSNs as well as monosynaptically coupled MNs and interneurons, on the ipsilateral side in relation to the injection, via the sensory neuron terminals. By contrast, no RabV labeled sensory or spinal neurons were observed in control experiments where we injected modified rabies virus in *RGT* mice lacking Cre (*Figure 6—figure supplement 1a–d*).

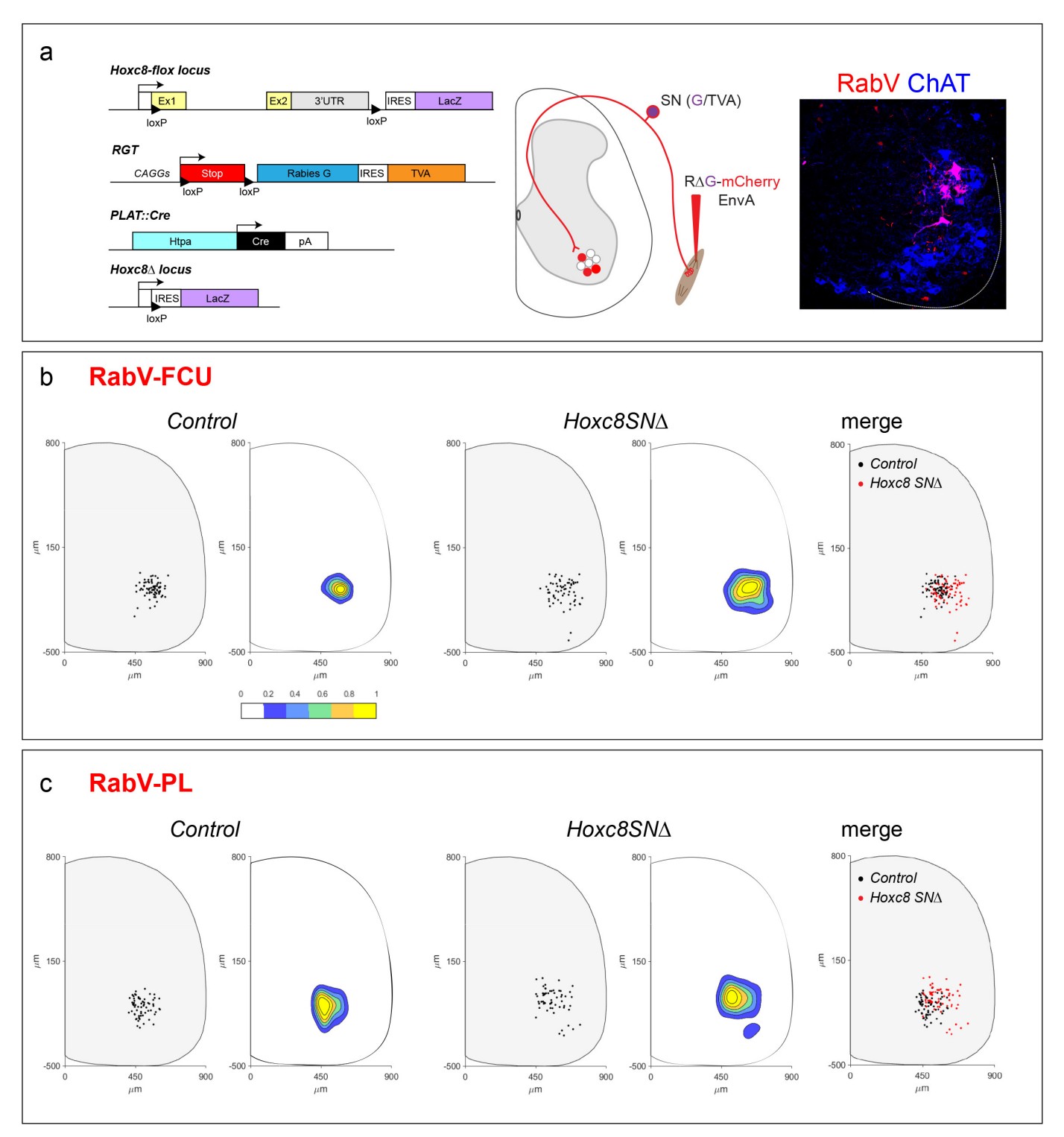

**Figure 6.** Altered central targeting of distal flexor muscle pSNs in *Hoxc8^{SNΔ}* mice. (**a**) Cre-dependent excision of *Hoxc8* coding sequence and expression of *LacZ* reporter driven by *PLAT* promoter. Removal of the stop cassette permits expression of 2 rabies helper proteins, TVA and glycoprotein G, after Cre-recombination thus directing rabies infection in a cell type specific manner (left). Modified rabies virus labeling experimental design: Env/TVA system permits primary infection of SNs; SNs also express glycoprotein G which enables monosynaptic transfer of injected modified rabies virus (RVΔG-mCherry-EnvA) anterogradely to spinal cord MNs by secondary infection (middle). Representative spinal cord cross-section of RVΔG-mCherry-EnvA monosynaptic labeling in MNs via pSN transfer after injection into FCU muscle of *PLAT::Cre; Hoxc8^{LacZ-flox/+}* mice (right). (**b and c**) Dot
*Figure 6 continued on next page*

Figure 6 continued

plot showing RVΔG-mCherry-EnvA labeled MNs position of a representative spinal cord hemi-section in the caudal cervical region. Distances from the central canal are shown on x and y coordinate axes (in micrometers). Contour plots to the right depicting labeled MN density in relation to position in spinal cord. Area of greatest labeling density in yellow. Overlay of labeled MN dot plots for both control and *Hoxc8$^{SNΔ}$* mice. Total number of labeled MNs in which the FCU is injected in control mice; N = 3 mice, 79 cells. Total number of labeled MNs in *Hoxc8$^{SNΔ}$* mice; N = 3 mice, 66 cells (b). Total number of labeled MNs in which the PL is injected in control mice; N = 3 mice, 71 cells. Total number of labeled MNs in *Hoxc8$^{SNΔ}$* mice; N = 3 mice, 62 cells. See also *Figure 6—figure supplement 1*.

The online version of this article includes the following figure supplement(s) for figure 6:

**Figure supplement 1.** Cre-dependent rabies infection of sensory neurons.

These results indicate that the rabies labeling of MNs is Cre-dependent and mediated through trans-synaptic spread via sensory central terminals.

We used this labeling strategy to map the overall distribution of postsynaptic targets of pSNs targeting a specific limb muscle in both control and *Hoxc8$^{SNΔ}$* RGT mice. We injected either the flexor carpi ulnaris (FCU) or the palmaris longus (PL) muscles with RabV, and mapped the location of trans-synaptically-labeled MNs, marked by Choline Acetyltransferase (ChAT). We then generated scatter plot and contour maps of the distribution of labeled RabV$^+$ ChAT$^+$ neurons (N = 3 animals). In control mice, injections into the FCU or PL labeled discrete clusters of RabV$^+$/ChAT$^+$ neurons located in a dorsal region of the caudal LMC (*Figure 6b,c*), consistent with the location of the MN pools targeting these muscles (*Bácskai et al., 2013*). By contrast, in *Hoxc8$^{SNΔ}$* mice rabies tracing from the FCU and PL labeled MNs that extended more ventrally and laterally within the LMC which of note, is typically the domain occupied by forelimb extensor MNs (*Figure 6b,c*). These qualitative observations suggest that *Hoxc8* regulates the pattern of pSN connectivity within the ventral spinal cord, presenting the possibility that flexor pSNs lacking *Hoxc8* may target inappropriate postsynaptic MN subtypes.

## Ectopic synapses between flexor pSNs and extensor MNs in *Hoxc8$^{SNΔ}$* mice

Because Hoxc8 expression is restricted to pSNs innervating distal forelimb flexor muscles, we next asked whether loss of *Hoxc8* leads to inappropriate synapses onto distal forelimb extensor MNs. To examine this, we injected distal flexor muscles with RabV, while concurrently retrogradely labeling MNs through injection of CTB into distal extensor muscles (*Figure 7a*). If removal of *Hoxc8* leads to an inappropriate coupling between flexor pSNs and extensor MNs, we would expect to observe colocalization of RabV$^+$ with CTB-labeled extensor MNs. Two distal extensor muscles were injected to maximize the possibility of finding ectopically connected MNs, and were also chosen based on their intrasegmental overlap with motor pools of injected flexors. To ensure no cross contamination of injected tracers, only superficial extensor muscles separated by at least three muscles from the injected flexors were chosen.

We injected a distal forelimb flexor muscle, FCU or PL, with RabV, and retrogradely labeled both distal forelimb extensor carpi radialis and extensor digitorum MNs with CTB. In control *PLAT::Cre$^+$ RGT* mice, the set of mCherry-labeled flexor MNs, labeled through transsynaptic viral spread via flexor pSNs, were distinct from retrogradely labeled CTB extensor MNs (control: N = 9 mice; FCU: N = 5, PL: N = 4) (*Figure 7b,d*). This result is consistent with electrophysiological and anatomical studies showing that the flexor pSNs do not synapse with extensor MN pools (*Eccles et al., 1957*; *Zampieri et al., 2014*).

By contrast, in *RGT Hoxc8$^{SNΔ}$* mice we observed ectopic connections originating from distal flexor pSNs onto distal extensor MNs (*Hoxc8$^{SNΔ}$*: N = 7 mice; FCU: N = 4, PL: N = 3) (*Figure 7b,d*). We quantified the fraction of MNs with coincident detection of RabV/CTB/ChAT over the total number of RabV labeled MNs in each injected animal. We found that in *Hoxc8$^{SNΔ}$* mice, in which the FCU is injected with rabies, 29 ± 5%, of the total RabV/ChAT labeled MNs colabeled with CTB (N = 4 mice), compared to 0 ± 0% in control animals (N = 5 mice, p<0.0001, Student's t-test) (*Figure 7c*). Similarly, in *Hoxc8$^{SNΔ}$* mice in which the PL was injected with rabies, 37 ± 12% of the total RabV/ChAT labeled MNs were CTB labeled (N = 3 mice), compared to that of 0.8 ± 0.8% in control animals (N = 4 mice, p=0.02, Student's t-test) (*Figure 7e*). The percentages likely underrepresent the entire cohort of

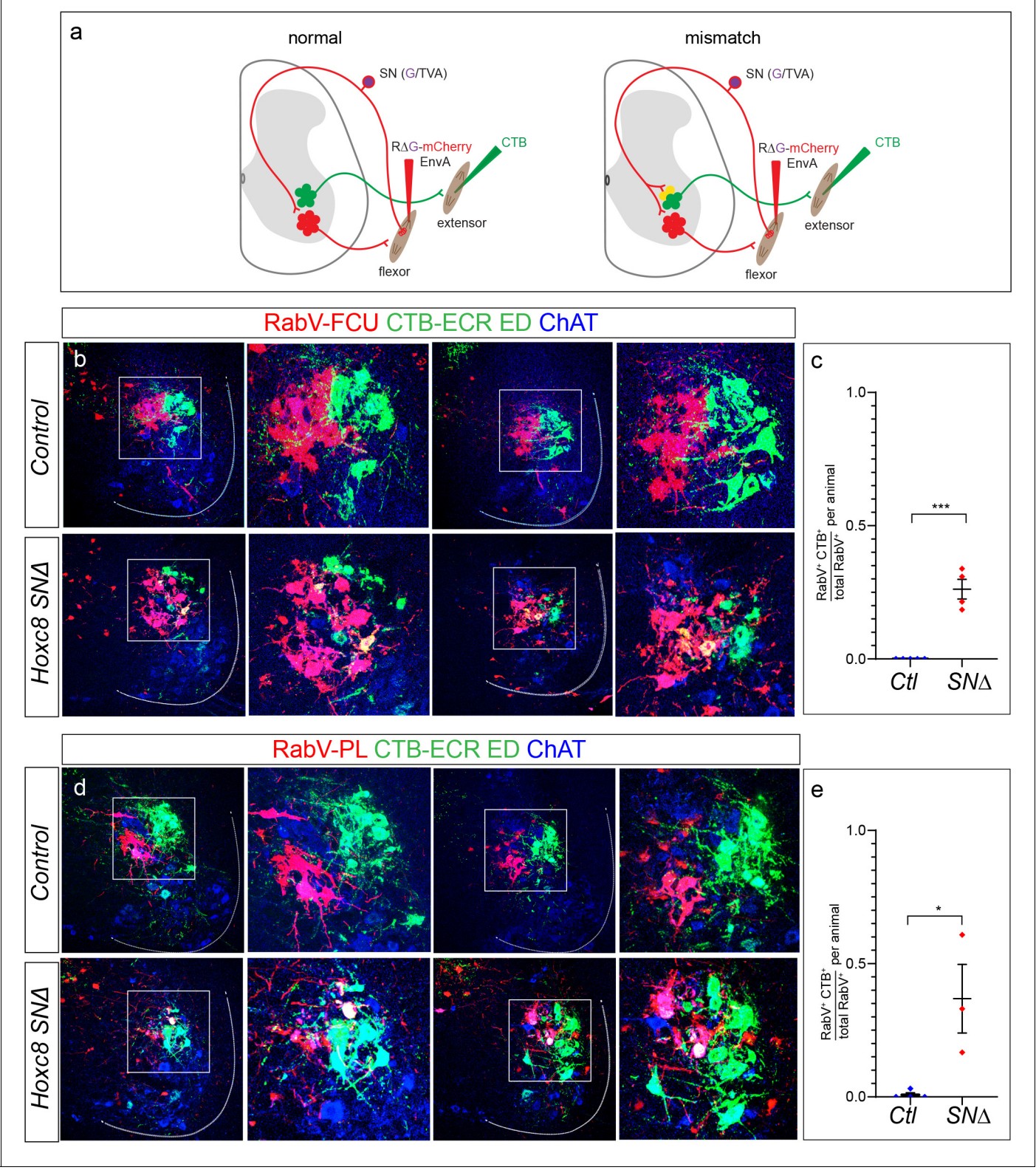

**Figure 7.** Flexor pSNs form ectopic synapses on extensor MNs in *Hoxc8^SNΔ* mice. (**a**) Schematic illustrating muscle injection of RVΔG-mCherry-EnvA (RabV). Selective infection of SNs and anterograde transsynaptic transport leads to secondary infection of MNs in the ventral spinal cord but not direct infection of MNs. CTB injected into a muscle labels directly connected MNs. In a normal condition, pSNs avoid synapsing onto antagonist MNs. In a mismatch condition, pSNs form ectopic contacts onto antagonist muscle MNs yielding colocalization of RabV (red) and CTB (green). Individual distal

*Figure 7 continued on next page*

*Figure 7 continued*

forelimb flexor muscles were injected with RabV and distal forelimb extensors were injected with CTB. Injections were performed at ~P6 P7 and spinal cords were collected at ~P12-13. (**b**) Colocalization of RabV, CTB, and ChAT signifying ectopic contacts in *Hoxc8$^{SN\Delta}$* mice where RabV was injected into the FCU and CTB was injected into the ECR and ED. (**c**) Quantification of the average percentage of RabV$^+$CTB$^+$ChAT$^+$ cells over total RabV$^+$ChAT$^+$ cells where RabV was injected into the FCU. Lines indicate mean ± SEM. Average from control mice: N = 5 mice; 0 ± 0. Average from *Hoxc8$^{SN\Delta}$*: N = 4 mice; 0.26 ± 0.04. (p<0.0001, Student's t test). (**d**) Colocalization of RabV, CTB, and ChAT signifying ectopic contacts in *Hoxc8$^{SN\Delta}$* mice where RabV was injected into the PL and CTB was injected into the ECR and ED. (**e**) Quantification of the average percentage of RabV$^+$CTB$^+$ChAT$^+$ cells over total RabV$^+$ChAT$^+$ cells where RabV was injected into the PL. Lines indicate mean ± SEM. Average from control mice: N = 4 mice; 0.008 ± 0.008. Average from *Hoxc8$^{SN\Delta}$*: N = 3 mice; 0.37 ± 0.13. (p=0.02, Student's t test). For both *Hoxc8$^{SN\Delta}$* mice and controls in which the FCU was injected, a total of ~150 RabV MNs were counted while 120 RabV MNs were counted for control mice injected in the PL and 101 RabV MNs for *Hoxc8$^{SN\Delta}$* mice. See also *Figure 7—figure supplement 1*.

The online version of this article includes the following source data and figure supplement(s) for figure 7:

**Source data 1.** Quantification of rabies labeled neurons in control and Hoxc8 mutants.
**Figure supplement 1.** Fidelity of sensory-motor specificity compromised is in *Hoxc8$^{SN\Delta}$* and *Hoxc8$^{MN\Delta}$* mice.

ectopically connected MN populations, since not all of the pSNs and thus MNs were infected by the rabies virus. Additionally, flexor pSNs ectopically contacted extensor MNs located within the same segments occupied by the flexor MNs, suggesting that flexor pSN projections mistarget within their normal rostrocaudal domains.

To confirm that ectopic synapses were formed by distal flexor pSNs onto distal extensor MNs, we employed a more conventional labeling strategy using CTB and the fluorescent tracer Rhodamine-dextran (Rh-Dex). After intramuscular injection, Rh-Dex is taken up by MNs, as well as pSN afferents, but is not transported transganglionically, thus restricting central tracing to MN soma. CTB, however, transfers into the central sensory axon, and accumulates in vGluT1$^+$ sensory boutons at the soma of synaptically-coupled MNs. Thus, after muscle injection we can compare pSN CTB labeling of vGlut1 synapses on Rh-Dex labeled MNs. We injected the FCU with CTB and distal extensor muscles with Rh-Dex in *RGT* control and *RGT Hoxc8$^{SN\Delta}$* mice. Similar to the results of the rabies tracing assay, we observed the presence of ectopic synapses from CTB/vGlut1 labeled distal flexor pSNs onto distal extensor Rh-Dex$^+$ MNs (*Figure 7—figure supplement 1a–c*). Together, these results indicate that *Hoxc8* plays an important role in pSNs during sensory-motor circuit development.

## Discussion

Animals rely on internal neural representations of muscle position and activity in order to execute coordinated motor behavior (*Akay et al., 2014*; *Mendes et al., 2013*; *Tuthill and Azim, 2018*). In vertebrates, pSNs establish selective central connections with MNs innervating the same peripheral muscle, while avoiding MNs targeting functionally antagonistic muscles. Whether pSNs are intrinsically programmed to acquire muscle-specific identities that enable them to target appropriate central postsynaptic targets is largely unknown. A major roadblock in resolving the mechanisms of spinal sensory-motor circuit assembly has been a lack of molecular tools to study muscle-specific pSN subtype differentiation. We found that pSNs innervating distal forelimb muscles can be defined by selective expression of Hox transcription factors, and these profiles are initiated independent of limb-derived cues. Additionally, *Hox* genes are critical in generating appropriate patterns of central connections between pSNs and MNs. We suggest that the coordinate activity of *Hox* genes in multiple neuronal classes plays a key role in establishing synaptic specificity within developing limb control circuits.

### *Hox* genes and sensory neuron diversification

Hox transcription factors are well known intrinsic determinants of patterning and cellular identities along the rostrocaudal axis of metazoans (*Mallo and Alonso, 2013*; *Philippidou and Dasen, 2013*). Our results indicate that, in addition to their broad roles in determining rostrocaudal positional identities, *Hox* genes have neuronal class-specific functions associated with the development of limb sensory-motor circuits. Our findings reveal that a subset of *Hox* genes are selectively expressed by pSNs generated at specific segmental levels, and these patterns parallel the rostrocaudal profiles of

*Hox* genes in the CNS. We found that the *Hoxc8* gene is preferentially expressed by pSNs targeting distal forelimb flexor muscles, important for wrist and digit movement, reflecting the Hoxc8 expression domain in MNs. While our studies focused on a single *Hox* gene, it is likely that other forelimb-specific pSN subtypes can be similarly delineated by specific combinations of *Hox4-Hox8* genes.

It is notable that pSNs express members of the *Hoxa* and *Hoxc* gene clusters, while *Hoxb* genes appear to be expressed by broader populations of sensory neurons, most of which are likely cutaneous. These patterns are reminiscent of the differential expression of *Hox* genes within the spinal cord, where *Hoxb* genes are typically expressed in dorsal populations, containing cutaneous sensory relay interneurons, while *Hoxa* and *Hoxc* genes are expressed by motor-related interneurons and MNs (*Dasen et al., 2005*; *Graham et al., 1991*; *Sweeney et al., 2018*). Dorsoventral differences in *Hox* patterning appear to emerge developmentally, as *Hox* transcripts are initially expressed uniformly in neural progenitors along the dorsoventral axis (*Liu et al., 2001*). While the mechanisms that govern the later bias of *Hoxa/c* and *Hoxb* expression in muscle and cutaneous sensory systems are unclear, they could relate to the timing of differentiation. In the spinal cord, ventral motor-related postmitotic neurons are born prior to dorsal types, and DRG neurons appear to exhibit a similar proprioceptive to cutaneous temporal progression (*Fariñas et al., 1996*; *Lawson and Biscoe, 1979*).

The dorsoventral restriction of genes within *Hox* clusters could provide a mechanism to diversify subtype identity across multiple sensory modalities. As cutaneous neurons are known to terminate in the dorsal spinal cord, the coordinate activities of *Hoxb* genes could similarly function in the development of cutaneous sensory-relay circuits. Consistent with this idea *Hoxb8* has been shown to be essential for normal development and organization of dorsal spinal interneurons, and loss of *Hoxb8* leads to excess grooming and reduced thermal and nociceptive response (*Holstege et al., 2008*).

## Extrinsic and intrinsic control of sensory-motor circuit development

Studies of sensory neuron development provide compelling evidence that a major determinant of subtype diversity and connectivity are instructive cues provided by peripheral limb muscle and mesenchyme. Expression of *Ntf3* within the developing limb regulates expression of *Etv1* and *Runx3* in pSNs, and differences in the levels of NT3 signaling contribute to muscle specific identities (*de Nooij et al., 2013*; *Wang et al., 2019*). The limb mesenchyme has also been implicated as a source of extrinsic cues which differentiate hindlimb extensor and flexor pSN subtypes (*Poliak et al., 2016*; *Wenner and Frank, 1995*). A confounding aspect in the study of limb-derived signaling in pSN development is that peripheral *Ntf3* signaling, as well as the intrinsic determinants *Etv1* and *Runx3*, are also required for sensory neuron survival, often precluding genetic analysis of later aspects of sensory-motor circuit development.

We found that *Hoxc8* is dispensable for pSN survival, and loss of *Hoxc8* does not prohibit the ability of pSNs to target their appropriate forelimb muscle targets. Moreover, expression of *Hox* genes in both forelimb-innervating pSNs and MNs is maintained in the absence of limb mesenchyme and muscle. While these results indicate a limb-independent mechanism of early neuronal differentiation, target-derived cues are likely required to establish the full molecular profiles of pSNs and functional characteristics. In spinal MNs, a major function of *Hoxc8* is to regulate expression of *Ret* and *Gfrα* genes, rendering a subset of MNs sensitive to activities of peripheral Gdnf to induce *Pea3* expression within motor pools (*Catela et al., 2016*). Thus, in MNs Hox proteins regulate expression of cell surface receptors that retrogradely influence subtype specification. *Hox* genes may similarly act in pSNs to modify or constrain the responses to peripheral cues as sensory axons navigate through the developing limb bud.

Expression of *Hoxc8* in pSNs is largely confined to distal forelimb flexor subtypes, while distal forelimb extensor pSNs lack *Hoxc8* pSN innervation. Interestingly, a recent study showed that *Runx3* is essential for the development of forelimb extensor pSNs, and suggests that this pattern is regulated by limb-derived cues (*Wang et al., 2019*). The differential expression of *Hoxc8* and *Runx3* in distal flexors and extensors could reflect refinement in the pattern of these factors by limb-derived cues. For example, Hoxc8 may antagonize Runx3 function within flexor pSN subtypes. Alternatively, limb-derived cues may maintain Runx3 in extensor pSNs and restrict *Hoxc8* expression to forelimb flexor pSNs.

### *Hox* genes and synaptic specificity in sensory-motor circuits

We found that *Hoxc8* is required in distal forelimb flexor pSNs to establish appropriate connections with their MN counterparts. How do *Hox* genes contribute to the specificity of central connections between pSNs and MNs? In spinal MNs, *Hox* genes and their downstream targets including *Pea3* and *Sema3e* have been shown to be essential for the specificity of their central connections with pSNs (*Baek et al., 2017*; *Pecho-Vrieseling et al., 2009*; *Vrieseling and Arber, 2006*), in part, by regulating MN topographical organization and dendritic architecture. In mice lacking the Hox accessory factor *Foxp1* the normal positioning of forelimb-innervating MNs is disrupted, leading to a sensory-motor mismatch (*Sürmeli et al., 2011*). The specificity of pSN-MN connections has been recently shown to correlate with the relative approach angles between pSN axons and MN dendrites (*Balaskas et al., 2019*), and loss of this alignment may account for the sensory-motor mismatch observed in both *Foxp1* and pSN *Hoxc8* mutants. Further studies will be necessary to definitively assess whether the coordinate regulation of MN and pSN connectivity by the same *Hox* gene contributes to sensory-motor specificity. Consistent with this model, in preliminary studies we found that after selective deletion of *Hoxc8* from MNs, MN pools are disorganized and distal forelimb flexor pSNs target extensor MNs (*Figure 5—figure supplement 1c,d*, *Figure 7—figure supplement 1d,e*). Although a Hox-specific molecular matching model for pSN-to-MN connectivity is provocative, *Hoxc8* could be required in pSNs, independent of *Hoxc8* in MN, for the segregation of axonal subtypes within the sensory nerve, or for their axon guidance within the spinal cord.

Centrally, pSNs establish connections with a variety of postsynaptic targets, including local spinal and projection interneurons that relay proprioceptive information to the brain (*Bermingham et al., 2001*; *Bikoff et al., 2016*; *Koch et al., 2017*; *Tripodi et al., 2011*; *Yuengert et al., 2015*). The same *Hox* genes expressed by pSNs and MNs are also expressed by multiple classes of spinal interneurons, suggesting a broader role in shaping synaptic specificity within the spinal cord. Consistent with this idea, both long ascending spinocerebellar and local-circuit spinal interneurons have been shown to require *Hox* function to acquire limb-specific molecular identities (*Baek et al., 2019*; *Sweeney et al., 2018*). Results from this work indicate that in addition to contributing to sensory neuron diversity, *Hox* genes are also required in pSNs to shape synaptic specificity in developing sensory-motor circuits. These observations are consistent with studies indicating that coordinate *Hox* activities are required in multiple neuronal and non-neuronal lineages during circuit assembly (*Barsh et al., 2017*; *Briscoe and Wilkinson, 2004*; *Zheng et al., 2015*). Our findings suggest the same *Hox* gene could act in multiple neuronal classes during development, implying a coherent molecular strategy for wiring the circuits essential for limb control.

## Materials and methods

### Key resources table

| Reagent type (species) or resource | Designation | Source or reference | Identifiers | Additional information |
|---|---|---|---|---|
| Genetic reagent (*M. musculus*) | *Hoxc8 flox* | PMID:19621436 | MGI: 4365797 | |
| Genetic reagent (*M. musculus*) | *PLAT::Cre* | PMID:12812797 | MGI: 3052515 | |
| Genetic reagent (*M. musculus*) | *Olig2::Cre* | PMID:18046410 | MGI: 3774124 | |
| Genetic reagent (*M. musculus*) | *Gt(ROSA)26Sor::CAG-loxp-STOP-loxp-rabies-G-IRES-TVA* | PMID:23352170 | MGI: J:206510 | |
| Biological sample (rabies virus) | *EnvA-RabV-mCherry* (pseudotyped G-deleted rabies virus) | PMID:21867879 PMID:17329205 PMID:26844832 | | ~$10^8$ IU/mL |
| Biological sample (chicken eggs) | SPF Eggs | Charles River | 10100332 | |
| Antibody | anti-Hoxc4 (Rabbit polyclonal) | PMID:16269338 | | (1:16000) |

*Continued on next page*

Continued

| Reagent type (species) or resource | Designation | Source or reference | Identifiers | Additional information |
|---|---|---|---|---|
| Antibody | anti-Hoxa5 (Rabbit polyclonal) | PMID:16269338 | | (1:16000) |
| Antibody | anti-Hoxc6 (Guinea pig polyclonal) | PMID:11754833 | RRID:AB_2665443 | (1:16000) |
| Antibody | anti-Hoxc6 (Rabbit polyclonal) | Aviva Systems Biology | Cat# ARP38484; RRID:AB_10866814 | (1:32000) |
| Antibody | anti-Hoxa7 (Guinea pig polyclonal) | PMID:16269338 | | (1:32000) |
| Antibody | anti-Hoxc8 (Mouse monoclonal) | Covance | RRID:AB_2028778 | (1:4000) |
| Antibody | anti-Hoxb4 (Rat monoclonal) | Developmental Studies Hybridoma Bank | Cat# I12; RRID:AB_2119288 | (1:100) |
| Antibody | anti-Hoxb5 (Rabbit polyclonal) | PMID:23103965 | | (1:32000) |
| Antibody | anti-Foxp1 (Rabbit polyclonal) | PMID:18662545 | RRID:AB_2631297 | (1:32000) |
| Antibody | anti-Isl1/2 (Mouse monoclonal) | Developmental Studies Hybridoma Bank | Cat# 39.4D5, RRID:AB_2314683 | (1:50) |
| Antibody | anti-Isl1/2 (Rabbit polyclonal) | Jessell lab | | (1:5000) |
| Antibody | anti-Meis2 (Rabbit polyclonal) | PMID:16269338 | | (1:16000) |
| Antibody | anti-Pbx3 (Rabbit polyclonal) | PMID:16269338 | | (1:16000) |
| Antibody | anti-Pea3 (Rabbit polyclonal) | PMID:9814709 | RRID:AB_2631446 | (1:32000) |
| Antibody | anti-βGal (Goat polyclonal) | Abcam | Cat# 9361; RRID:AB_307210 | (1:1000) |
| Antibody | anti-βGal (Goat polyclonal) | Santa Cruz | Cat# sc-19119; RRID:AB_2111604 | (1:2000) |
| Antibody | anti-Ret (Goat polyclonal) | Santa Cruz | Cat# sc-1290; RRID:AB_631316 | (1:100) |
| Antibody | anti-CTB (Goat polyclonal) | List Biological Laboratories | Cat# 703; RRID:AB_10013220 | (1:4000) |
| Antibody | anti-vGlut1 (Guinea pig polyclonal) | Millipore | Cat# AB5905; RRID:AB_2238022 | (1:1000) |
| Antibody | anti-PV (Rabbit polyclonal) | Swant | Cat# PV27; RRID:AB_2631173 | (1:1000) |
| Antibody | anti-Runx3 (Rabbit polyclonal) | Abcam | Cat# ab68938; RRID:AB_1141661 | (1:16000) |
| Antibody | anti-TRITC (Rabbit polyclonal) | Thermofisher | Cat# A6397; RRID:AB_2536196 | (1:1000) |
| Antibody | anti-Ret (Rabbit polyclonal) | Cell Signaling | Cat# 3223; RRID:AB_2238465 | (1:100) |
| Antibody | anti-ChAT (Rabbit polyclonal) | Jessell Lab | | (1:16000) |
| Antibody | anti-Etv1 (Rabbit polyclonal) | Jessell Lab | | (1:8000) |
| Antibody | anti-βGal (Chick polyclonal) | Jessell Lab | | (1:5000) |
| Antibody | anti-cRunx3 (Guinea pig polyclonal) | Jessell Lab | | (1:5000) |
| Antibody | Alexa 488-, Cy3-, Alexa 647- secondaries (Donkey polyclonal) | Jackson Immuno Research | | (1:1000) |
| Peptide, recombinant protein | 1% Cholera Toxin B subunit | Sigma-Aldrich | Cat# C9903 | |
| Chemical compound, drug | Dextran, Tetramethyl - rhodamine | ThermoFisher | Cat# D3308 | |
| Software, algorithm | Fiji | PMID:22743772 | RRID:SCR_002285 | http://imagej.net/Fiji |
| Software, algorithm | Imaris | Bitplane/Oxford Instruments | v8.1.2 RRID:SCR_007370 | |

| Reagent type (species) or resource | Designation | Source or reference | Identifiers | Additional information |
|---|---|---|---|---|
| Software, algorithm | Matlab | Mathworks | R2019b RRID:SCR_001622 | |
| Software, algorithm | GraphPad Prism | GraphPad Software | 8.0.2 (263) | |

## Mouse genetics

*PLAT::Cre* (*Pietri et al., 2003*), *Hoxc8 flox* (*Blackburn et al., 2009*), *Olig2::Cre* (*Dessaud et al., 2007*), and *RGT* (*Takatoh et al., 2013*) mouse lines have been previously described. Cre-floxed systems for targeted gene deletion were used to obtain genotypes of interest. Breeding combinations of *PLAT::Cre* with *Hoxc8 flox* were performed to generate conditional heterozygous mice (*Hoxc8$^{LacZ-flox/+}$*) as well as homozygous mutant mice (*Hoxc8$^{SNΔ}$*). *Hoxc8$^{LacZ-flox/+}$* and Cre negative *Hoxc8 flox* mice were used as controls. No gross phenotypic differences between *Hoxc8$^{LacZ-flox/+}$*, *Hoxc8 flox* and wild-type mice were observed. Breeding combinations of homozygous *RGT* mice were mated with homozygous *Hoxc8 flox* mice to obtain *Hoxc8 flox-RGT* mice. *Hoxc8$^{SNΔ}$* and *Hoxc8$^{LacZ-flox/+}$* were then bred with *Hoxc8 flox-RGT* to generate mice heterozygous for both *Hoxc8 flox* and *RGT, Hoxc8$^{SNΔ}$-RGT$^{flox/+}$* as well as *Hoxc8$^{LacZ-flox/+}$-RGT$^{flox/+}$*. *Hoxc8 flox* mice were crossed with *Olig2::Cre* mice to generate MN-specific mutants (*Hoxc8$^{MNΔ}$* mice). Animal work was approved by the Institutional Animal Care and Use Committee of the NYU School of Medicine in accordance with NIH guidelines.

## Immunohistochemistry

Mouse embryos were harvested between E11.5 to E18, fixed in 4% paraformaldehyde (PFA) for 1.5–2 hr, cryoprotected in 30% sucrose/PBS overnight and cryosectioned at 10 um or 16 um thickness. Postnatal day P4-P10 mice were perfused with PBS and 4% PFA and post-fixed for 2 hr at 4C prior to cryoprotection and cryosectioning. Primary antibodies against Hox proteins, Foxp1, Isl1/2, Meis2 and Pea3 have been previously described (*Dasen et al., 2005*; *Jung et al., 2010*). Additional antibodies used: Hoxc8-Alexa 488 (mouse, 1:2000, Covance), Hoxc6 (rabbit, 1:32000, Aviva Systems Biology), βGal (goat, 1:2,000, Santa Cruz), Ret (goat, 1:100, Santa Cruz), CTB (goat, 1:4,000, List Biological Laboratories), vGluT1 (guinea pig, 1:1,000, Millipore), PV (rabbit, 1:1000, Swant), Runx3 (rabbit, 1:16,000, Abcam), TRITC (rabbit, 1:1,000, ThermoFisher), Ret (rabbit, 1:100, Cell Signaling), ChAT (rabbit, 1:16,000, Jessell lab), Etv1 (rabbit, 1:8000, Jessell lab), βGal (chicken, 1:5,000, Jessell lab), βGal (chicken, 1:1,000, Abcam), Islet1/2 (mouse, 1:50, Developmental Studies), Pbx3 (guinea pig, 1:16,000, Dasen lab), cRunx (guinea pig, 1:5,000, Jessell lab). Detailed protocols for histology are available on the J.S.D. lab website (http://www.med.nyu.edu/dasenlab/).

## Chick limb ablations

Unilateral limb ablations were performed between stages 16–18 (*Calderó et al., 1998*) and embryos incubated to develop to stages 26–28. Embryos were sacrificed and further processed once full limb ablation was confirmed. Spinal cords with attached DRG were dissected and immersed in 4%PFA for 1–2 hr at 4C followed by cryprotection in 30% sucrose overnight. Tissue was cryosectioned at 16 um.

## Muscle extraction

Mice were sacrificed at P12 by transcardial perfusion and whole muscle dissections were performed with the animal preparation submerged in ice cold 1X PBS solution. Following removal, each muscle was pinned down in a sylgard plate and immersed in 4%PFA for 2 hr at 4C followed by cryoprotection in 30% sucrose solution overnight. Muscles were embedded in mounting media and cryosectioned at 16 um thickness.

## Virus production

Local stocks of virus were used to amplify, purify, and concentrate rabies virus (RVΔG-mCherry-EnvA) according to established protocols (*Osakada and Callaway, 2013*; *Wickersham et al., 2007*). RVΔG-mcherry virus was produced and amplified in B7GG cells and subsequently pseudotyped with EnvA in BHK-EnvA cells to produce RVΔG-mCherry-EnvA with minor modifications to protocol. After BHK-EnvA cells were infected,~7 hr later, cells were washed three times in PBS and fresh medium was added, and this was repeated the following day. After a subsequent 48 hr incubation, medium was harvested, filter purified, and viral particles were concentrated by ultracentrifugation. Concentrated virus was then resuspended in PBS and viral titer was assessed by serial dilution of the virus on HEK293t cells to achieve a viral titer of ~$1 \times 10^8$ IU/mL.

## Tracing experiments

### Sensory neuron labeling

1% CTB (Sigma-Aldrich) solutions were injected with a glass capillary into a forelimb muscle of anesthetized mice at P4-P5 and examined after 3 days. Pups were perfused with PBS and 4% PFA. Spinal cords with attached DRG were isolated and post-fixed for 2–3 hr at 4C. Tissue was cryosectioned at 16 um following cryoprotection.

### Sensory and motor neuron labeling

For anterograde labeling of sensory and motor neurons,~0.8 uL of RVΔG-mCherry-EnvA virus was injected with a glass capillary into either the FCU or PL of anesthetized mice at P5-P6 which were then perfused with PBS and 4% PFA 5 days later. Spinal cords were isolated, processed, and cryosectioned at 16 um or 30 um. Muscle injection specificity was verified post-mortem by the exclusive presence of fluorescent labeling in muscle of interest.

### Double labeling

To anterogradely label sensory neurons and monosynaptically connected motor neurons, RVΔG-mCherry-EnvA virus was injected into either the FCU or PL of mice at ~P5, as described above, for analysis of ectopically labeled motor neuron soma. Concurrently,~10–50 nl of 1% CTB solution was injected into the ECR and ED muscles. Animals were perfused with PBS and 4% PFA 5 days later. Spinal cords were isolated, processed, and cryosectioned at 16 um.

To anterogradely label sensory neuron synapses onto motor neurons,~30–50 nl of 1% CTB solution was injected into either the FCU or PL muscle of ~P5 mice, as described above. Concurrently, TMR-Dextran (Rh-Dex) was injected into the ECR and ED muscles to retrogradely label motor neurons. Animals were perfused with PBS and 4% PFA ~3 days later. Spinal cords were isolated, processed, and cryosectioned at 30 um.

## Quantification and statistical analyses

### Neuronal cell counts and muscle spindle analysis

Neuronal cell counts were performed on 10 or 16 um cryosections obtained from caudal cervical DRG. Images were acquired using an LSM 700 Zeiss confocal microscope and cell counts were calculated using the Fiji/ImageJ cell counter feature. For chick limb ablation assays, neuronal cell counts were compared between DRG of the ablated and non-ablated sides of an individual chick embryo. Neuronal cell counts in which forelimb muscles were injected with 1% CTB were performed on 16 um cryosections from caudal cervical DRG and processed/analyzed as described above. Quantifications were done based on comparable labeling efficiency between all injected animals for each muscle type and was required to have a minimum of 10 PV$^+$ CTB-labeled sensory neurons. Tissue sections of 16 um thick muscle tissue were imaged to analyze muscle spindle projections. Sensory endings within muscle spindles were identified based on the presence of vGluT1$^+$ terminals with characteristic annulospiral morphology. While each unique spindle of an entire muscle was not counted, a series of sections of the whole muscle was profiled to obtain a representative sample of muscle spindles for each muscle. For both cell counts and spindle analysis, serial sections throughout the entire tissue sample were collected into 8 and 5 parallel series of sections respectively and at least one full series of sections was compared between controls and mutants or limb ablated chick

embryos. Analyses were performed on N $\geq$ 3 mice/genotype or per muscle type and N = 3 limb ablated chick embryos.

## Quantification of pSN collateral density (PV fiber density)

Quantitative analysis of pSN fiber density in the ventrolateral region of the spinal cord was performed on collapsed confocal Z-stacks using Fiji/ImageJ analysis software. The total PV+ collateral area (calculated as the mean pixel intensity) was measured within a confined lateral region of the ventral spinal cord at the segmental level of DRG C8 set by the borders of the midline and the ventral and lateral gray matter and white matter boundaries. An ROI was set to cover the ventrolateral region to be quantified and measured 45,832 pixels. The threshold was designated based on PV labeling only in the ventrolateral area and the same threshold value was used across all animals. For each genotype, N = 3 animals were analyzed.

## Quantitative analysis of motor neuron position

Plotting of labeled motor populations was performed on 30 um cryosections. Tiled images were acquired with a Zeiss confocal microscope (LSM 700) at 10X. X-Y coordinates for motor neuron soma measured in um units were determined with respect to the central canal using IMARIS software. Contour plots were generated from the X-Y scatter plots and six isolines were automatically assigned in Matlab.

## Quantitative analysis of ectopic motor neuron soma

RV$\Delta$G-mCherry-EnvA/CTB/ChAT labeled motor neurons were analyzed from 16 um serial cryosections of the cervical spinal cord. Coincident labeling of soma was quantified using the cell counter feature in ImageJ. For each genotype/forelimb flexor muscle type, at least N = 3 animals were analyzed. Only animals with comparably efficient labeling were used for analysis. Efficient labeling was designated as a minimum of 20 CTB/ChAT$^+$ MNs and 10 RabV/ChAT$^+$ MNs. Serial sections throughout the entire tissue sample were collected into 10 parallel series of sections and three full series of sections were compared for each animal.

## Analysis of sensory synaptic contacts with motor neurons

Analysis of vGluT1$^+$ sensory bouton contacts with P7–P9 motor neuron soma and ~100 μm proximal dendritic arbor was performed using 0.4 μm confocal z stacks of 30 μm thick sections using a 63X oil objective lens. Gamma-motor neurons were excluded from analysis. Distance of boutons on dendritic arbor from soma was assessed using the scale bar set by Zen software. Images were analyzed using Fiji/ImageJ.

### Statistics

Samples sizes were determined based on previous experience and the number of animals and definitions of N are indicated in the main text and figure legends. In figures where a single representative image is shown, results are representative of at least two independent experiments, unless otherwise noted. No power analysis was employed, but sample sizes are comparable to those typically used in the field. Data collection and analysis were not blind. Graphs of quantitative data are plotted as means with standard error of mean (SEM) as error bars, using Prism 8 (Graphpad) software. Significance was determined using unpaired (Student's) t-test and was calculated using Prism eight software. Exact p values are indicated in the main text and figure legends.

### Acknowledgements

We thank Kristen D'Elia, Joriene de Nooij, and David Schoppik for comments on the manuscript, and Niels Ringstad, Jim Salzer, and Jessica Treisman for valuable feedback. We also thank for Myungin Baek, Sara Fenstermacher, Alana Mendelsohn, Polyxeni Philippidou, Rocio Rivera and NYU Langone's Microscopy Laboratory for technical assistance and preliminary data. This work was supported by NIH NINDS grants T32 NS086750 to MS, R35 NS116858, R01 NS062822 and R01 NS097550 to JD.

## Additional information

### Funding

| Funder | Grant reference number | Author |
|---|---|---|
| National Institutes of Health | R35 NS116858 | Jeremy Dasen |
| National Institutes of Health | R01 NS097550 | Jeremy Dasen |
| National Institutes of Health | T32 NS086750 | Maggie M Shin |
| National Institutes of Health | R01 NS062822 | Jeremy Dasen |

The funders had no role in study design, data collection and interpretation, or the decision to submit the work for publication.

### Author contributions

Maggie M Shin, Conceptualization, Formal analysis, Funding acquisition, Investigation, Methodology, Writing - original draft, Writing - review and editing; Catarina Catela, Conceptualization, Formal analysis, Investigation, Methodology, Writing - original draft, Writing - review and editing; Jeremy Dasen, Conceptualization, Funding acquisition, Investigation, Methodology, Writing - original draft, Writing - review and editing

### Author ORCIDs

Maggie M Shin  https://orcid.org/0000-0001-7891-0774
Jeremy Dasen  https://orcid.org/0000-0002-9434-874X

### Ethics

Animal experimentation: This study was performed in strict accordance with the recommendations in the Guide for the Care and Use of Laboratory Animals of the National Institutes of Health. All of the animals were handled according to an approved institutional animal care and use committee (IACUC) protocol (IA16-00045) of the NYU School of Medicine.

### Decision letter and Author response

Decision letter https://doi.org/10.7554/eLife.56374.sa1
Author response https://doi.org/10.7554/eLife.56374.sa2

## Additional files

### Supplementary files

• Transparent reporting form

### Data availability

All data generated or analysed during this study are included in the manuscript and supporting files.

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
