## [Decision Letter]

**Acceptance summary:**

The paper presents a new model of extrinsic and intrinsic control of sensory-motor circuit development, and how *hox* genes regulate synaptic specificity in sensory-motor circuits. We think that *eLife* readers will be interested in the findings, and we thank the authors for their detailed attention to the reviewer criticisms, most importantly the addition of quantification of the results for the benefit of our readers.

**Decision letter after peer review:**

Thank you for submitting your article "Intrinsic control of neuronal diversity and synaptic specificity in a proprioceptive circuit" for consideration by *eLife*. Your article has been reviewed by two peer reviewers, and the evaluation has been overseen by a Reviewing Editor and Marianne Bronner as the Senior Editor. The reviewers have opted to remain anonymous.

The reviewers have discussed the reviews with one another and the Reviewing Editor has drafted this decision to help you prepare a revised submission.

Summary:

The role of *Hox* genes in sensory-motor circuit assembly is largely unknown. The role for pSN *Hox* codes in synaptic specificity has been poorly explored because 1) *Hox* deletion needs to be conditional to prevent MN defects or lethality, and 2) pSNs innervating different muscles are intermingled inside a dorsal root ganglion (DRG), making it difficult to individually study synaptic targeting of each pSN subtype. This is a careful study addressing intrinsic mechanisms of neuronal specificity and the role of *Hox* transcription factors circuit generation-in this case matching proprioceptive sensory neurons to their targets both in muscle and spinal cord neurons. The authors have overcome these challenges by utilizing a Cre line specific to spinal sensory neurons, and also by introducing a new anterograde labeling technique. This virus-mediated labeling approach is particularly remarkable because this enables selective visualization of entire MNs that a given sensory subtype synapses on, despite their sparse distribution inside a DRG. With this strategy, the authors successfully demonstrated that *Hoxc8* is required for caudal pSNs to form specific monosynaptic connections with their appropriate MN targets. This study demonstrates that *Hoxc8* in subsets of dorsal root ganglia neurons is not required for survival of the neurons, matching to muscles, or extending to the appropriate region within the spinal cord. But *Hoxc8* is required in the sensory neurons for precisely targeting the motor neurons innervating limb flexor muscles, and in its absence, the sensory neuron forms inappropriate synapses on motor neurons for limb extensor muscles. These findings are convincingly shown and carefully reported. Although this is shown for one *Hox* factor, this may have broader impact as a mechanism for sensory-motor matching. Results from this study suggest that the coordinated regulation of the same *Hox* gene in pSN and MNs is essential for the specificity of connections in proprioceptive circuits. This paper suggests a novel molecular code that defines muscle- specific proprioceptive sensory neurons subtypes, shows that proprioceptive sensory neurons subtype identities are intrinsically programmed in early development, and that *Hox* genes are required for proprioceptive sensory neurons central afferents to target the appropriate postsynaptic targets within the spinal cord. In terms of significance the work is appropriate for publication in *ELife*.

Both reviewers had misgivings about the work. For example, in its current form, reviewers point out many places where there is insufficient data to support the conclusions that are being made. Also, where the data is sufficient to make a conclusion, other interpretations that could be equally valid and are not ruled out.

Essential revisions:

1) While the most challenging experiments in the paper (Figures 4, 6 and 7) are quantified, many of the foundational experiments in the earlier figures are represented by a single image, with no n's and consequently no statistical analysis. In some cases, these are key negative findings, such as in the limb bud extirpation experiment in Figure 2 and Figure 2—figure supplement 1 (see below) or the very important finding that conditional knockout of *Hoxc8* in sensory neurons doesn't alter pSN identity (Figure 5E – see below).

2) The important positive result of the altered localization of pSN synapses on motor neurons after conditional KO of *Hoxc8* in SNs or MNs in Figure 7—figure supplement 2 is presented anecdotally. In some cases where quantitative data has been obtained, such as in Figure 6, it is not determined whether differences are statistically significant. Although the intrinsic *Hox* code model for sensorimotor circuit formation is an appealing one and the experiments are perfectly designed to test it, the lack of quantitation of much of the data presented in the paper make it difficult to assess whether the model is supported.

3) The first few figures describe the expression of the *Hox* family factors in the dorsal root ganglia. This is important for setting up ideas on how they make function. However, as presented, it is not clear which *Hox* factors were actually assessed (it says Hox4-8-is this all clusters? Were some not looked at?) and what the characteristics are for each? The Materials and methods section doesn't help either because it just states the antibodies for the *Hox* factors have been described. This information needs to be more precise. We suggest a table summarizing all the Hox factors tested and a description of the characteristics would be valuable (supplement is fine). These should include the points made in the first few figures like is it expressed in DRG and does it overlap with markers of proprioceptive identity, what is the rostral caudal restriction if any, which *Hox* factors are co-expressed.

4) Limb bud removal experiments: In arguing for an autonomous *Hox* program in pSNs, this paper contradicts previous reports showing that muscle-derived cues specify pSN identity and connectivity with MNs. The key finding in the current paper supporting an intrinsic mechanism is the chick limb bud extirpation experiments in Figure 2 and Figure 2—figure supplement 1. However, the conclusions need to be better supported. While Figure 2I gives the sense that *Hoxc8* expression is unaffected after limb bud removal, Figure 2—figure supplement 1C reveals that only a small fraction of *Hoxc8*-expressing neurons is in fact pSNs (this is different from mouse). If most of the *Hoxc8*-expressing sensory neurons are cutaneous, it is expected that they would be unaffected by removal of limb bud muscles. Figure 2—figure supplement 1C is thus the more important finding if it shows that the number of *Hoxc8*-expressing pSNs is unaffected after limb bud ablation. However, the number of *Hoxc8^+^Runx3^+^* neurons is small (2-3) in both images. More n's and quantification of *Hox^+^Runx3^+^* neurons are needed to support the conclusion that pSN *Hox* expression is unaffected by removal of the limb bud.

5) Figure 1—figure supplement 1A: schematizes *Hox* expression in DRGs along the rostro-caudal axis but the data only shows where the various *Hox* genes are expressed, not where they are not expressed, so we have to trust that the schematic accurately represents data that is not shown. All the rostro-caudal levels that are included in the schematic should be shown for all four *Hox* genes, as these authors did very beautifully in Catela et al., 2016 Figure 1.

6) Figure 5E is an important figure showing that pSN identity is unaffected by conditional KO of *Hoxc8*. Rather than being presented as a single figure with three or four co-expressing neurons, this experiment should be done in *Hoxc8* SN∆ mutants as quantitatively as the wildtype data presented in Figure 4.

7) Based on the finding in Figure 7 showing that *Hoxc8* mutant pSNs inappropriately target extensor-innervating motor neurons, the authors conclude that a *Hox* matching mechanism required for synapse formation between pSNs and their appropriate MN targets has been disrupted. While this is an appealing possibility, other interpretations are equally possible. For instance, *Hoxc8* may be important for the segregation of pSN axon subtypes in the sensory nerve, or for their axon guidance within the spinal cord. A prediction of the *Hox* matching mechanism is that flexor-innervating pSNs will fail to synapse on flexor-innervating MNs in *Hoxc8* MN∆ animals and may inappropriately innervate extensor-innervating MNs. A single example of the latter result is shown (Figure 7—figure supplement 2D, E) – more examples of this outcome would strengthen the *Hox* matching model.

---

## [Author Response]

Essential revisions:1) While the most challenging experiments in the paper (Figures 4, 6 and 7) are quantified, many of the foundational experiments in the earlier figures are represented by a single image, with no n's and consequently no statistical analysis. In some cases, these are key negative findings, such as in the limb bud extirpation experiment in Figure 2 and Figure 2—figure supplement 1 (see below) or the very important finding that conditional knockout of Hoxc8 in sensory neurons doesn't alter pSN identity (Figure 5E – see below).

We have addressed each of these issues with additional quantification, further analyses, and provision of specific information on: 1) the number of animals analyzed for each experiment, including those in which a single representative image is shown, 2) clarifying the specific statistical methods used in the paper, and 3) further quantifying the results presented in Figures 2, Figure 2—figure supplement 1, and Figure 5. New quantifications are now plotted in Figures 2F, J, K, L, Figure 5F and Figure 2—figure supplement 1D. See details below.

2) The important positive result of the altered localization of pSN synapses on motor neurons after conditional KO of Hoxc8 in SNs or MNs in Figure 7—figure supplement 2 is presented anecdotally. In some cases where quantitative data has been obtained, such as in Figure 6, it is not determined whether differences are statistically significant. Although the intrinsic Hox code model for sensorimotor circuit formation is an appealing one and the experiments are perfectly designed to test it, the lack of quantitation of much of the data presented in the paper make it difficult to assess whether the model is supported.

In the revision we now provide a more thorough quantification of key experimental data, added additional statistical analyses, and provided specific information on the statistical methods used. We have quantified the results shown in Figure 2 (see point 4 below) and Figure 5F (point 6 below). The data shown in Figure 6 is intended to be qualitative, in showing mislocalization of rabies-labeled neurons in the *Hoxc8* SN mutant. The corresponding quantitative results, which required retrograde labeling of extensor MNs is shown in Figure 7C, E. We have clarified this in the revision. We have also revised our presentation of the data shown in Figure 7—figure supplement 2, and have limited our claims in the paper with respect to a pSN-MN “molecular matching” model (see point 7 below).

3) The first few figures describe the expression of the Hox family factors in the dorsal root ganglia. This is important for setting up ideas on how they make function. However, as presented, it is not clear which Hox factors were actually assessed (it says Hox4-8-is this all clusters? Were some not looked at?) and what the characteristics are for each? The Materials and methods section doesn't help either because it just states the antibodies for the Hox factors have been described. This information needs to be more precise. We suggest a table summarizing all the Hox factors tested and a description of the characteristics would be valuable (supplement is fine). These should include the points made in the first few figures like is it expressed in DRG and does it overlap with markers of proprioceptive identity, what is the rostral caudal restriction if any, which Hox factors are co-expressed.

We thank the reviewers for this excellent suggestion and apologize for neglecting to include this important information. In the revision we have added a table indicating specifically which *Hox* genes were tested as well as those that were untested. We have included in this table: 1) whether the *Hox* protein is expressed in SNs, 2) its rostrocaudal domain of expression, and where possible, 3) whether any other *Hox* protein is coexpressed with the *Hox* protein under consideration (this is constrained by the host species of certain *Hox* antibodies).

We have also clarified in the main text that not all *Hox4-Hox8* paralog genes were analyzed. We have added precise information on the *Hox* antibody reagents used in the Key Resources Table.

4) Limb bud removal experiments: In arguing for an autonomous Hox program in pSNs, this paper contradicts previous reports showing that muscle-derived cues specify pSN identity and connectivity with MNs. The key finding in the current paper supporting an intrinsic mechanism is the chick limb bud extirpation experiments in Figure 2 and Figure 2—figure supplement 1. However, the conclusions need to be better supported. While Figure 2I gives the sense that Hoxc8 expression is unaffected after limb bud removal, Figure 2—figure supplement 1C reveals that only a small fraction of Hoxc8-expressing neurons is in fact pSNs (this is different from mouse). If most of the Hoxc8-expressing sensory neurons are cutaneous, it is expected that they would be unaffected by removal of limb bud muscles. Figure 2—figure supplement 1C is thus the more important finding if it shows that the number of Hoxc8-expressing pSNs is unaffected after limb bud ablation. However, the number of Hoxc8^+^ Runx3^+^ neurons is small (2-3) in both images. More n's and quantification of Hox^+^Runx3^+^ neurons are needed to support the conclusion that pSN Hox expression is unaffected by removal of the limb bud.

In the revision we have quantified the fraction of all Isl1^+^ sensory neurons that express each of the three *Hox* proteins we analyzed in these experiments, both in controls and after forelimb ablation. We find that the fraction of SNs expressing *Hox* proteins is unchanged after limb bud ablation. However, consistent with recent studies (Wang et al., 2019), the number of SNs expressing the proprioceptive marker *Runx3* is reduced by ~50% (new Figure 2—figure supplement 1D)- making it difficult to assess the fraction of pSNs that maintain *Hox* expression. It has also been previously demonstrated that the pSN-class specific marker Er81 is markedly reduced after limb bud ablation (Lin et al., 1998). Nevertheless, while the fraction of SNs that express *Hoxc8* is relatively small in chick relative to mouse, we find that this percentage is unchanged after limb ablation (new Figure 2J-L). We added these results to main text as follows:

“The fraction of Isl1^+^ SNs expressing *Hoxa5* (44.3 ± 4.7% in controls, 50.7 ± 3.2% ablated, p=0.27, Student’s t-test), *Hoxc6* (26.8 ± 2.0% controls, 26.3 ± 3.0% ablated, p=0.90), and *Hoxc8* (6.4 ± 0.9% controls, 7.6 ± 0.8% ablated, p=0.32) was not significantly changed (Figure 2J, K, L).”

We would also like to note that these ablation experiments not only remove limb muscle, but all limb-derived tissues (e.g. limb mesenchyme) that might provide instructive signals to both muscle and cutaneous SN subtypes. As the reviewers are likely aware, recently studies from the Ginty lab (Sharma et al., 2020) provide evidence that target-derived cues play important roles in specifying the transcriptional identity of both cutaneous and proprioceptive SNs. Thus, while *Hoxc8* is expressed by both cutaneous and proprioceptive SNs in chick, this pattern also appears to be target-independent.

We would also like to clarify that our study in no way contradicts the important role that target-derived cues play in pSN differentiation and connectivity. Our results provide evidence for a crucial limb-independent and *Hox*-dependent program that operates in conjunction with limb-dependent signaling programs. We have modified our Discussion to highlight this:

“While these results indicate a limb-independent mechanism of early neuronal differentiation, target-derived cues are *undoubtedly* required to establish the full molecular profile of pSNs and their functional characteristics.”

5) Figure 1—figure supplement 1A: schematizes Hox expression in DRGs along the rostro-caudal axis but the data only shows where the various Hox genes are expressed, not where they are not expressed, so we have to trust that the schematic accurately represents data that is not shown. All the rostro-caudal levels that are included in the schematic should be shown for all four Hox genes, as these authors did very beautifully in Catela et al., 2016 Figure 1.

The results in Figure 1I show a top-down view of *Hoxa5* and *Hoxc8* expression in DRG and spinal cord, which includes DRG that lack expression of these proteins. In the revision we have added top-down expression analyses for *Hoxc4*, *Hoxc6*, and *Hoxa7* in Figure 1—figure supplement 1A that similarly show the DRG lacking these proteins.

Unfortunately, we did not collect the negative *Hox* expression data for the serial sections shown in Figure 1—figure supplement 1A. Nevertheless, the data shown in Figure 1—figure supplement 1A were taken from a single embryo in which the relative rostrocaudal position for each section was recorded for each *Hox* antibody stain. Thus, the relative positioning of the rostrocaudal domains for *Hoxc4* and *Hoxc6* are accurate with respect to *Hoxa5* and *Hoxc8* in these panels, but only approximate relative to segmental level (i.e. C1-T1). We have clarified this in the figure legend, and indicate that similar results were obtained from a total of N=3 embryos analyzed at this stage.

6) Figure 5E is an important figure showing that pSN identity is unaffected by conditional KO of Hoxc8. Rather than being presented as a single figure with three or four co-expressing neurons, this experiment should be done in Hoxc8 SN∆ mutants as quantitatively as the wildtype data presented in Figure 4.

We thank the reviewer for this suggestion. In the revision we have quantified the results from the retrograde tracing experiments and now present them in Figure 5F and the main text, as follows:

“We injected CTB into the FCU of *Hoxc8^SN∆^* mice at P4 and collected spinal cords with attached DRG at P7. We found that the fraction of pSNs that were βGal^+^ CTB^+^ was similar between controls and *Hoxc8^SN∆^*mice (65.19 ± 0.05% for N=4 control animals; 62.27 ± 0.08% for N=3 *Hoxc8^SN∆^*mice, p=0.75, Student’s t-test) (Figure 5E, F).”

7) Based on the finding in Figure 7 showing that Hoxc8 mutant pSNs inappropriately target extensor-innervating motor neurons, the authors conclude that a Hox matching mechanism required for synapse formation between pSNs and their appropriate MN targets has been disrupted. While this is an appealing possibility, other interpretations are equally possible. For instance, Hoxc8 may be important for the segregation of pSN axon subtypes in the sensory nerve, or for their axon guidance within the spinal cord. A prediction of the Hox matching mechanism is that flexor-innervating pSNs will fail to synapse on flexor-innervating MNs in Hoxc8 MN∆ animals and may inappropriately innervate extensor-innervating MNs. A single example of the latter result is shown (Figure 7—figure supplement 2D, E) – more examples of this outcome would strengthen the Hox matching model.

In the revision, we have reworded our interpretation of these results, and only suggest a “matching model” as one possibility in the Discussion. We have also included other interpretations of the results as the reviewers have thoughtfully suggested. Unfortunately, due to technical constraints, we cannot perform the rabies-mediated transynaptic assays on the *Hoxc8* MN mutant, which would enable a more rigorous quantitative and definitive assessment of synaptic specificity deficits, as we were able to do with the SN-restricted knockout. We are also currently unable to repeat the synaptic tracing assay, due to the effects of the Covid19 shutdown on our mouse colony. We have therefore moved the *Hoxc8* MN mutant analysis from the main results to the Discussion, and explicitly state that analysis of *Hoxc8* MN mutants are preliminary and require additional supporting data. This section now reads as follows:

“Further studies will be necessary to definitively assess whether the coordinate regulation of MN and pSN connectivity by the same *Hox* gene contributes to sensory-motor specificity. […] Although a *Hox*-specific molecular matching model for pSN-to-MN connectivity is provocative, *Hoxc8* could be required in pSNs, independent of *Hoxc8* in MN, for the segregation of axonal subtypes within the sensory nerve, or for their axon guidance within the spinal cord.”

We believe the idea that *Hox* genes are needed in MNs during sensory-motor connectivity is additionally supported by the analyses of the *Hox* accessory factor *Foxp1* (Surmeli et al., 2011) and MN-*Hoxc9* mutants (Baek et al., 2017). We agree it remains to be determined whether there is an exact matching between muscle-specific pSNs and MN pools subtypes, but feel it is not an overstatement to say *Hox* genes are needed in both neuronal classes during sensory-motor circuit development.